# Mouse V1 population correlates of visual detection rely on heterogeneity within neuronal response patterns

Jorrit S Montijn[1]*, Pieter M Goltstein[1,2], Cyriel MA Pennartz[1,3]*

[1]Swammerdam Institute for Life Sciences, Center for Neuroscience, Faculty of Science, University of Amsterdam, Amsterdam, Netherlands; [2]Max Planck Institute of Neurobiology, Martinsried, Germany; [3]Research Priority Program Brain and Cognition, University of Amsterdam, Amsterdam, Netherlands

**Abstract** Previous studies have demonstrated the importance of the primary sensory cortex for the detection, discrimination, and awareness of visual stimuli, but it is unknown how neuronal populations in this area process detected and undetected stimuli differently. Critical differences may reside in the mean strength of responses to visual stimuli, as reflected in bulk signals detectable in functional magnetic resonance imaging, electro-encephalogram, or magnetoencephalography studies, or may be more subtly composed of differentiated activity of individual sensory neurons. Quantifying single-cell $Ca^{2+}$ responses to visual stimuli recorded with in vivo two-photon imaging, we found that visual detection correlates more strongly with population response heterogeneity rather than overall response strength. Moreover, neuronal populations showed consistencies in activation patterns across temporally spaced trials in association with hit responses, but not during nondetections. Contrary to models relying on temporally stable networks or bulk signaling, these results suggest that detection depends on transient differentiation in neuronal activity within cortical populations.

*For correspondence: j.s. montijn@uva.nl (JSM); c.m.a. pennartz@uva.nl (CMP)

**Competing interests:** The authors declare that no competing interests exist.

## Introduction

Lesion studies in humans and animals indicate the causal importance of the primary visual cortex (V1) in detection, discrimination, and awareness of visual stimuli (*Lashley, 1943*; *Weiskrantz et al., 1974*; *Weiskrantz, 1996*), and this role has been recently confirmed by direct optogenetic inhibition of mouse V1 (*Glickfeld et al., 2013*). Visual perception has been proposed to arise from interactions between stimulus-specific processing in V1 and neural activity in higher visual and frontoparietal areas, involving both feed-forward propagation of activity and recurrent, top-down feedback (*Shadlen and Newsome, 1996*; *Britten and van Wezel, 1998*; *Lamme et al., 2000*; *Haynes et al., 2005*). Critical in unraveling neural correlates of vision is how detected and undetected stimuli are processed differently, especially when these stimuli are physically identical. For instance, it has been suggested that the intensity, duration, and reproducibility of sensory neural activity may provide signatures critical for visual perception (e.g. *Moutoussis and Zeki, 2002*; *Schurger et al., 2010*). In addition, it has been proposed that neural activity in V1 does not correlate with visual perception because stimuli that were seen or not seen evoked similar V1 blood-oxygenation-level-dependent signals (*Vuilleumier et al., 2001*; *Rees, 2000*), but this remains an area of substantial controversy (*Ress and Heeger, 2003*; *Palmer et al., 2007*; *Nienborg and Cumming, 2014*). In this context, it is important to recall that functional magnetic resonance imaging (fMRI), electro-encephalogram (EEG), and magnetoencephalography (MEG) rely on a mean-field approach, leaving open the

**eLife digest** Seeing is not the same as perceiving, where an object is recognized and information about it is interpreted by the brain. Things might be in your field of view, but not actively perceived; for example, when daydreaming with your eyes open. Many researchers have investigated how the brain responds differently to a perceived object compared with something that is seen but not perceived. However, using relatively coarse techniques, only small differences in brain activity have been found.

Many of the techniques used to investigate brain activity only look at the average activity of a group of neurons – the cells in the brain that process information. This raises the possibility that the perception of an object relies on more subtle or complex interactions in brain activity. To investigate this, Montijn et al. trained mice to lick a reward spout that gave out sugar water when they perceived a particular image. A technique called two-photon calcium imaging was then used to simultaneously record the activity of tens to hundreds of neurons in part of the brain called the visual cortex as the mice performed the perception task.

This revealed that the average activation of a group of neurons was only weakly related to whether a mouse had perceived the image. However, differences in the strength of the responses of the individual neurons in the group reflected perception more strongly: when a mouse perceived the image and licked in response, a heterogeneous (non-uniform) set of neuronal responses occurred. The diversity of the neuronal responses could also be used to predict how quickly a mouse would respond to an image. These activity differences would not be picked up by techniques that detect the average activity of many neurons, explaining why these effects had not previously been seen.

These findings shed light on which patterns of activity in the visual region of the brain lead to objects being perceived or not. Whether similar mechanisms operate in different regions of the brain remains to be investigated.

possibility that neural correlates of perception may be coded in more subtle ways that take into account the local differentiation present in populations of sensory neurons.

Such local, functional differentiation is supported by single- or multiunit recording studies in visual, auditory, and somatosensory areas of animals trained to make perceptual decisions (*Logothetis et al., 1995*; *Britten et al., 1996*; *Posner and Gilbert, 1999*; *Petersen, 2002*; *Luna et al., 2005*; *Palmer et al., 2007*; *Mitchell et al., 2009*; *Cohen and Maunsell, 2009*; *Cohen and Maunsell, 2011*; *Sachidhanandam et al., 2013*; *Chen et al., 2013*; *Miyashita and Feldman, 2013*; *Doron et al., 2014*; *McGinley et al., 2015*). Over the last decade, it has become clear that the shared response variability between neurons (i.e. noise correlation) might be particularly important for sensory processing because noise correlations can influence the amount of information that can be extracted from neuronal population codes (*Averbeck et al., 2006*; *Cafaro and Rieke, 2010*). Furthermore, it has been observed that these correlations can be reduced during stimulus presentation (*Gutnisky and Dragoi, 2008*; *Snyder et al., 2014*) and directed attention, which may aid in disentangling stimulus information from noisy population responses (*Mitchell et al., 2009*; *Cohen and Maunsell, 2009*; *Herrero et al., 2013*).

Although noise correlations have been studied well, they have the drawback of not being an instantaneous measure—their computation requires integrating neural activity over multiple time points or stimulus repetitions. Instantaneous aspects of population activity in cortex, such as temporal spike co-occurrence and population sparseness, seem critical for efficient neural coding (*Olshausen and Field, 1997*; *Vinje, 2000*; *Benucci et al., 2013*; *Harris and Mrsic-Flogel, 2013*). Some population-based measures have been proposed and tested in somatosensory and auditory cortex (*Romo et al., 2003*; *Safaai et al., 2013*; *Carnevale et al., 2013*; *Buran et al., 2014*). It has, for example, been shown that measures based on the variability and correlations between neurons correlate better with the animal's decision than simpler approaches based on the mean spiking rate (*Safaai et al., 2013*; *Carnevale et al., 2013*). However, in the domain of visual perception the behavioral relevance of only few population measures has been experimentally tested in paradigms where animals report behaviorally whether they have seen a stimulus or not.

Therefore, we investigated correlates of visual stimulus detection using two-photon calcium imaging of populations of ~100 neurons in V1 L2/3 of mice performing a detection task as superficial layers are easy to access with calcium imaging and have been reported to show neural correlates with stimulus detection (*van der Togt, 2006*; *Ito and Gilbert, 1999*). Our first aim was to examine whether visual detection correlates with the mean visual response strength of V1 neurons or rather with other metrics of population responses, such as noise correlation or variance. This led us to develop a novel population metric—response heterogeneity—that correlates better with stimulus detection performance, and particularly with the animal's reaction time, than traditional measures by capturing the dissimilarity of neuronal responses within a population. Second, an assumption in many computational models of vision is that neurons in distributed cortical architectures have relatively fixed roles in encoding visual features, but modulate their activation in a temporally dynamic manner based on attentional needs that can influence perception (e.g. *Jones and Palmer, 1987*; *Itti et al., 1998*; *Desimone, 1998*; *Dayan and Abbott, 2001*; *Deco and Rolls, 2004*; *Reynolds and Heeger, 2009*). To study whether modulations of neuronal activity that influence stimulus perception show temporally recurring patterns, we asked whether population activation patterns are more similar across trials that repeat the same stimulus presentation when the stimulus is successfully detected. We report that (1) visual stimulus detection does not correlate well with mean response strength, but is significantly correlated with population heterogeneity; (2) neuronal populations show consistencies in activation patterns across temporally spaced trials in association with hit responses, but not when the animal fails to report a stimulus; and (3) in addition to heterogeneity, multidimensional structures in neuronal population responses provide information on visual detection.

## Results

To investigate how ensembles of primary visual cortex (V1) neurons are involved in visual detection, we trained mice to perform a go/no-go stimulus detection task (*Figure 1a*). After task acquisition, we performed two-photon calcium imaging in V1 contralateral to the visually stimulated eye (*Figure 1—figure supplement 1*, animals were awake, head-fixed, and performed a detection task where they indicated by licking whether a square-wave drifting grating was presented. Stimulus duration was delimited by the onset of the first licking response, with a maximum of 3.0 s for no response; therefore, no licks occurred during presentation of the stimulus. To acquire a sufficient range of hit/miss ratios, we presented test stimuli with different luminance contrasts: 0.5%, 2%, 8%, and 32%. These test trials were interleaved with 0% no-contrast and 100% full-contrast probe trials to estimate the animals' ratio of false alarms and omissions due to lack of motivation. For all analyses, we discarded trials where animals responded within 150 ms after stimulus onset (0.3–3.5% of trials per animal) because such fast responses may be ascribed to spontaneous licking.

To quantify behavioral performance during execution of the task, we calculated the 2.5th–97.5th percentile intervals [henceforth 95% confidence intervals (CIs)] of response proportions to the two types of probe trials: no-contrast and full-contrast stimuli. All eight animals (see 'Materials and methods') showed a significantly above-chance visual detection of square-wave drifting gratings during the acquisition of neural data (*Figure 1c*) (non-overlapping Clopper–Pearson 95% CIs). Behavioral response proportions increased with higher stimulus contrasts (*Figure 1e*) (group-level linear regression analysis, p<0.001) and mean reaction times decreased (*Figure 1f*) (p<0.005).

### Response dissimilarity within neuronal populations correlates with detection

As a first approach to examine population correlates of visual detection, we investigated differences in mean activity levels between hit and miss trials (*Figure 2*). We defined each neuron's response during a trial as the mean dF/$F_0$ during the entire stimulus presentation (*Figure 2a,b*). Because hit/miss would arguably be stronger in the population of neurons that prefer the features of the visual stimulus, we started out with investigating neural correlates of detection in the preferred population. We, therefore, calculated each neuron's preferred stimulus orientation (see 'Materials and methods'), and for the analysis in *Figure 2c,d* took for each trial the responses of only neurons that preferred the presented stimulus orientation (henceforth 'preferred population'). In *Figure 2c*, all trials of a single animal were grouped by stimulus contrast and behavioral response [hit/false-alarm ('response') or miss/correct-rejection ('no-response')], and the average preferred population

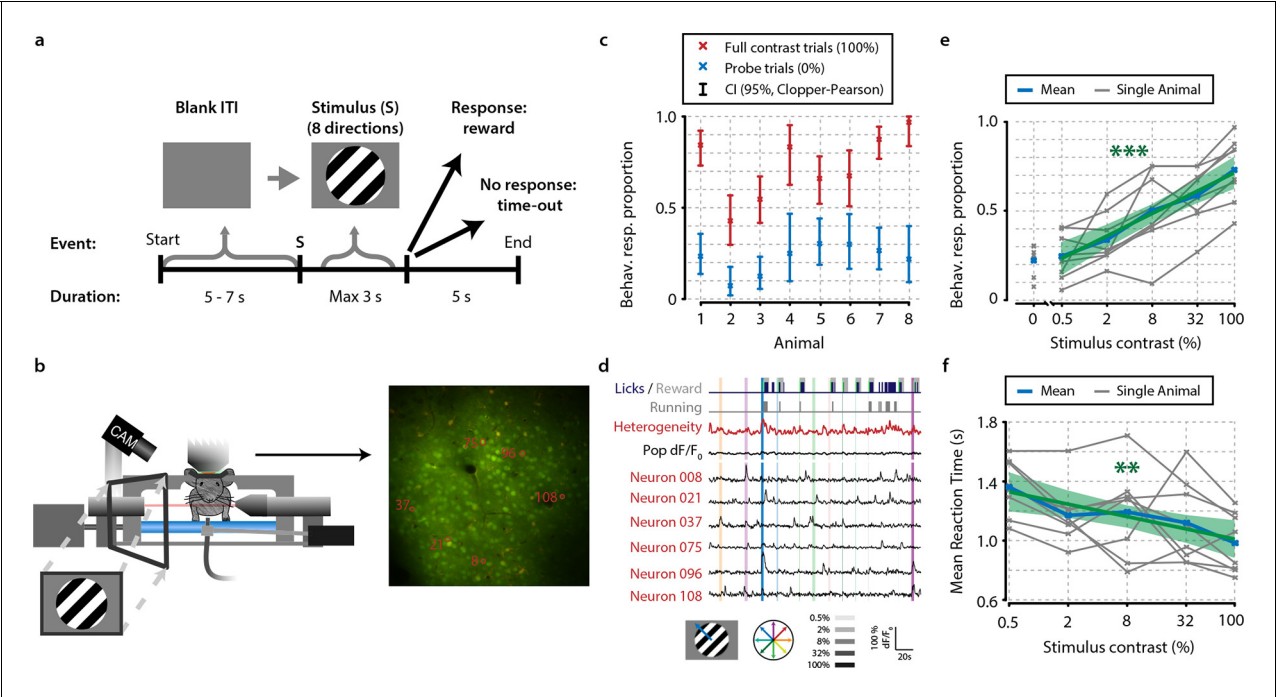

**Figure 1.** Mice perform a go/no-go task during in vivo calcium imaging. (**a**) Task schematic showing the time course of a single trial. In each trial, one of a combination of eight different directions and five contrasts, or a 0% contrast probe trial (isoluminant gray blank screen) was presented (ratio 1:5 of 0%-contrast-probe:stimulus trials). When mice made a licking response during stimulus presentation, the visual stimulus was turned off and sugar water was presented. (**b**) Schematic of experimental setup. During task performance, we recorded eye movements with an infrared-sensitive camera, licking responses, and running on a treadmill. (**c**) All eight animals performed statistically significant stimulus detection during neural recordings, as quantified by non-overlapping 2.5th–97.5th Clopper–Pearson (CP) percentile confidence intervals (95% CI) ($p<0.05$) of behavioral response proportions for 0% and 100% contrast probe trials. (**d**) Example of simultaneously recorded behavioral measures, population heterogeneity, mean population $dF/F_0$, and traces of neurons labeled in panel (**b**) Vertical colored bars represent stimulus presentations; width, color, and saturation represent duration, orientation, and contrast, respectively. (**e, f**) Animals showed significant increases in behavioral response (behav. resp.) proportion (linear regression analysis, see 'Materials and methods', $p<0.001$) (**e**) and reductions in reaction time ($p<0.01$); (**f**) with higher stimulus contrasts. Shaded areas show the standard error of the mean. Statistical significance: **$p<0.01$; ***$p<0.001$.

The following figure supplement is available for figure 1:

**Figure supplement 1.** Neuronal signals are stable over time.

response was calculated for hits and misses. As expected, the mean response increased with higher stimulus contrasts (*Figure 2—figure supplement 1* for traces across time). However, for this animal we did not find a significant difference between hit and miss trials for any individual contrast [false-discovery rate (FDR)-corrected paired t-test, $p>0.05$ for all contrasts] (note that for both false alarms and correct rejections V1 mean population response was indistinguishable from zero; *Figure 2c,d*, 0% contrast; *Figure 2—figure supplement 1a*). When grouping the test contrasts (0.5–32%), the data did show a modestly higher response for hit than miss trials for single animals as well as across animals ($p<0.05$). We, therefore, asked whether this increase in neuronal responses during stimulus detection was due to consistent response enhancements of specific neurons or due to a population-distributed process.

Including again also nonpreferred neurons for all further analyses (unless stated otherwise), we calculated the hit modulation ($dF/F_0$ increase during hits relative to misses) per neuron per hit trial (see 'Materials and methods') (*Figure 2e*(I)) and investigated whether this hit modulation could be explained by a subgroup of neurons that consistently enhances its activation during detection trials, random trial-by-trial population fluctuations, or both (*Figure 2e*). Hit modulation was explained to a small but significant extent by neuronal identity [$R^2=0.059$; $p<0.05$; *Figure 2e,f*], and to a larger extent by population fluctuations across trials [$R^2=0.248$; *Figure 2e*(III)] or both processes together

[$R^2=0.281$; *Figure 2e*(IV)]. The number of consistently hit-modulated neurons (which could be either up- or downregulated) was significantly above chance (*Figure 2f*; p<0.05). This pattern was robust over animals as hit modulations could be explained with above-chance accuracy by neuron identity, population fluctuations, or both in 7/8, 8/8, and 8/8 animals, respectively (*Figure 2g*). The fraction of significantly hit-modulated neurons was above chance (at p<0.05) for 7/8 animals. Although significant for most subjects, the variance explained by the consistency of neuronal responses was fairly low (always $R^2<0.1$), and even the combination of trial-by-trial population fluctuations and neuronal identity never exceeded $R^2=0.35$. This could indicate either that detection-related neural correlates in V1 are minor or that a simple enhancement of mean activity is an index ill-suited to describe potentially strong, but more complex changes in neuronal population dynamics. In particular, we hypothesized that correlates of stimulus detection may be unfolding by multineuron interactions at the single trial level and rely on the relative contrast in activation between neurons.

Several metrics aim to quantify response heterogeneity within neuronal populations, such as the sparseness (*Field, 1994*), or variance (*Seung and Sompolinsky, 1993*). However, such metrics are rarely studied in the context of behavioral relevance, and in the few cases where they are, their ability to predict behavior appeared modest (*Froudarakis et al., 2014*). Therefore, we developed an alternative measure of population heterogeneity that aims to capture the spread in normalized population activity (*Figure 3a,b*; see also 'Materials and methods'): by subtracting the z-scored response (each trial being a single data point per neuron, see *Equation 2*) of each neuron from that of all other neurons in that same trial, we obtained a Δz-score matrix where high values indicate high pairwise dissimilarity in neuronal activation. Taking the mean over all pairwise Δz-scores provides a measure of population heterogeneity that can in theory be computed over an arbitrarily small time interval (but note that for all analyses, except those shown in *Figure 5*, we used a single trial as time unit). This way, similarly strongly activated as well as similarly weakly activated pairs of neurons will decrease heterogeneity. By contrast, dissimilarly activated neuronal pairs (i.e. one strong, one weak) will increase it. Therefore, population heterogeneity incorporates both trial-by-trial fluctuations and intra-population differences in a neuronal pairwise manner. Its dependence on z-scored activity means that a neuron's contribution to heterogeneity is scaled to its relative level of activation—and because highly active neurons are often highly variable (*Baddeley et al., 1997*; *Montijn et al., 2014*) also to its signal-to-noise ratio.

We applied this metric to the activity of neurons from the entire population during hit and miss trials and found a stronger correlation with behavioral stimulus detection than for mean response strength (see *Video 1* and *Figure 3c* for a single animal example). Test contrasts (0.5–32%) showed a highly significant overall increase in heterogeneity for hit trials (*Figure 3c*, paired t-test, p<0.001), but such modulations were absent for probe trials. This difference was consistent over animals (*Figure 3d*) and showed similar patterns for the within-preferred and within-non-preferred population heterogeneity (*Figure 3—figure supplement 1*). Using a measure of effect size over animals (Cohen's *d*), we observed that heterogeneity showed a stronger correlation with visual detection than mean $dF/F_0$ (*Figure 3f*; three paired t-tests vs. $dF/F_0$ were all p<0.05). Linear single-trial prediction of hit or miss responses with a receiver operating characteristic (ROC) analysis on either mean $dF/F_0$ or heterogeneity showed that behavioral responses could be predicted above chance at single-trial basis with both metrics, but heterogeneity showed a significantly higher prediction score [area under curve (AUC), t-test across animals, $dF/F_0$ vs. 0.5; p<0.05, heterogeneity vs. 0.5; p<0.001, $dF/F_0$ vs. heterogeneity; p<0.01] (*Figure 3g,h*). These results show that correlates of visual detection are better captured by the strength of pairwise response dissimilarities within the neuronal population than to overall increases in mean activation (but note that correlation-based measures also work well for hit–miss differentiation; *Figure 3—figure supplement 2*).

## Heterogeneity predicts reaction time

Our observations suggest that not a general gain increase in population activity, but rather more complex changes in response strengths within a population determine the behavioral accuracy. Behavioral reaction time is often used as a proxy for salience, attention, and readiness (*Beck et al., 2008*), so we hypothesized that similar dissociations for fast/slow responses may be found as for hit/miss trials. We performed linear regressions per animal for $dF/F_0$ (*Figure 4a*) and heterogeneity (*Figure 4d*) as a function of reaction time. Similarly to hit/miss differences, the preferred population $dF/F_0$ was not significantly associated with behavioral performance (regression slopes vs. 0, FDR-

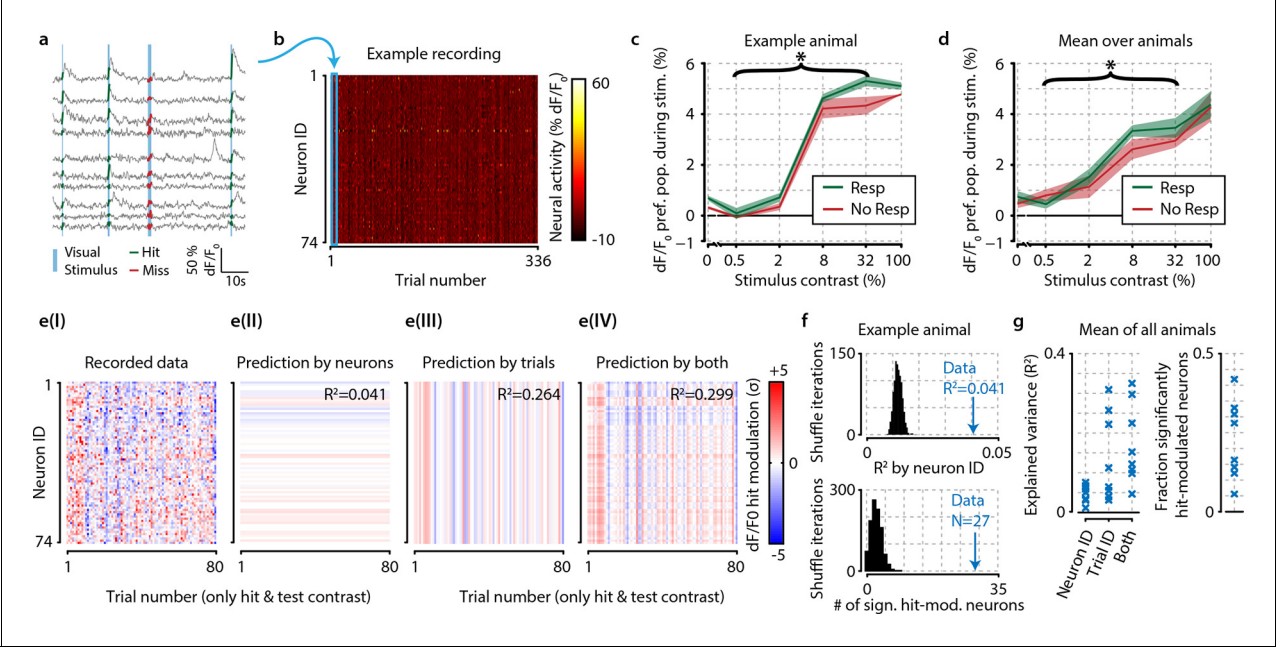

**Figure 2.** The difference in neural activity between hit and miss trials can be partly explained by consistent hit-associated increases in activity of specific neurons, and somewhat better by trial-by-trial population-wide fluctuations, but these mean-based approaches fall short of being fully descriptive. (**a**) Recorded traces from 10 randomly selected neurons over four subsequent trials. For further analysis, we took the mean $dF/F_0$ per neuron over the visual stimulation period (thick colored lines) as single mean neural activity measure per trial. (**b**) Data of one entire recording block consisting of 74 tuned and simultaneously recorded neurons over 336 trials. Blue rectangle shows the four trials depicted in panel (**a**). (**c**) In an example animal, the detection of stimuli (green) with test contrasts (0.5–32%) correlated with a modest increase in preferred population $dF/F_0$ over undetected stimuli (red) (two-sample t-test, $p < 0.05$) but none of the individual contrasts reached statistical significance (resp. vs. no resp., two-sample t-tests, FDR-corrected $p > 0.05$). (**d**) As (**c**), but for mean over all animals the graph shows a small, but consistent overall difference of $dF/F_0$ with visual detection (test contrasts 0.5–32%, n=8 animals, $p < 0.05$). (**c, d**) Shaded areas show the standard error of the mean. (**e**) The hit-associated increase in neural activity per neuron for all hit trials of test contrast stimuli (panel I) can be partly explained by specific neurons showing consistent $dF/F_0$ increases or decreases across trials (panel II) partly by trial-by-trial population-wide fluctuations regardless of neuronal identity (panel III) and somewhat better by both (panel IV). (**f**) Control analysis by shuffling neuronal identities (IDs) per trial (n=1000 iterations, black distribution) shows that the population activity is more predictable based on consistent hit modulations per neuron (top panel) and more neurons are significantly hit-modulated (bottom panel) than can be expected by chance. (**g**) Analyses as in (**f**), but across animals; comparison versus shuffle-based $R^2$-expectation showed above-chance (at $\alpha = 0.05$) predictability of hit modulations using neuron ID, trial ID or both for respectively 7/8, 8/8, and 8/8 animals (left panel). The fraction of significantly hit-modulated neurons was above chance (at $\alpha = 0.05$) for 7/8 animals (right panel).

The following figure supplement is available for figure 2:

**Figure supplement 1.** Several control analyses reveal no confounding effects of motor preparation, modulatory feedback, and eye movements.

corrected one-sample t-test, n.s.), nor were preferred population z-scored activity, variance, sparse-ness, instantaneous Pearson-like correlations (see 'Materials and methods'), whole-population (raw and z-scored) $dF/F_0$, and sliding-window based correlations (*Figure 4—figure supplement 1*). How-ever, heterogeneity and the spread in instantaneous Pearson-like correlations were inversely corre-lated with reaction time ($p < 0.001$, $p < 0.01$, respectively) and explained significantly more reaction-time-dependent variance in the data than all other measures (FDR-corrected pairwise t-tests, hetero-geneity vs. all, $p < 0.05$). This relationship holds when analyzed over animals (*Figure 4c*) as well as per individual animal (*Figure 4—figure supplement 1*).

Our definition of heterogeneity is computationally somewhat similar to the width of the distribu-tion of pairwise neuronal correlations in a population (see 'Materials and methods'). However, whereas the spread in instantaneous Pearson-like correlations is based on multiplying z-scored pair-wise neuronal responses and taking their standard deviation, the heterogeneity metric (which instead uses the mean absolute distance in z-scored $dF/F_0$ between pairs of neurons) was an even better predictor of behavioral reaction times (*Figure 4c*, $p < 0.05$). As such, it is more closely related to the

population mean of nondirectional neuron-pairwise Mahalanobis (i.e. normalized Euclidian) distances than Pearson's correlations. Our analysis shows that visual detection correlates well with large mean Mahalanobis distances in neural activity between pairs of neurons; that is, a high heterogeneity in population activity.

Nonetheless, it could be argued that changes in population activity might be uncorrelated with the fidelity with which the population code represents visual stimuli. To address this, we used a Bayesian maximum-likelihood decoder to assess the presence of a stimulus from V1 population activity (see also 'Materials and methods'; *Montijn et al., 2014*). Decoding performance was higher for behaviorally correct detection trials (*Figure 4d*; hit vs. miss, p<0.05), and the performance was similar to the animals' actual behavioral performance at a global level across contrasts (shuffled vs. non-shuffled, p<0.001; *Figure 4e*), as well as at a single-trial level (chi-square similarity analysis of hit/ miss trials for behavioral response and stimulus presence decoding, $\chi^2$=135.36, p<$10^{-30}$; *Figure 4— figure supplement 2*). Moreover, additional analyses revealed that stimulus features (orientation, contrast) were better decodable when the animal made a correct detection (*Figure 4—figure supplement 2a*) and when heterogeneity was high (*Figure 4—figure supplement 2b–d*). Thus, stimulus features, such as orientation, are represented more accurately by neuronal populations in V1 during hit trials, even though the specific orientation was irrelevant for the animal to perform the visual stimulus detection task.

During higher levels of arousal, it has been observed that neuronal activation is more desynchronized (*Cohen and Maunsell, 2009*; *Froudarakis et al., 2014*). Based on our current observations, this led us to hypothesize that a high heterogeneity in V1 populations reflects a brain state conducive to stimulus detection. If correct, heterogeneity immediately prior to stimulus presentation should be predictive of reaction time. To test this, we split all hit trials into the slowest 50% and fastest 50% per contrast (e.g. see *Figure 5a–d*) and calculated a measure of predictability of slow versus fast responses based on the 3 s preceding stimulus presentation (*Figure 5—figure supplement 1*). Using pre-stimulus-onset heterogeneity, fast response trials were highly predictable (FDR-corrected one-sample t-tests, p<0.01), while slow versus miss trials were not (p=0.799). Behavioral responses were not predictable based on population dF/$F_0$ (slow-miss, p=0.157; slow-fast, p=0.811; fast-miss, p=0.924), and the difference in predictability between heterogeneity and dF/$F_0$ was significant for slow-fast (p<0.01) and fast-miss (p<0.05), but not for slow-miss (p=0.477) trials.

Although heterogeneity before stimulus onset thus predicts behavior, we also found a dissociation between detected (slow and fast responses) and undetected stimuli (miss trials) in the rise time latency to maximum heterogeneity upon stimulus onset (p<0.05). Detected trials correlated with fast rise times, while neuronal response heterogeneity to undetected stimuli ramped up much more slowly (*Figure 5f*). This argues against the interpretation that heterogeneity merely reflects a tonic brain state that can be fully gauged before stimulus onset. The formation of nonhomogenous response patterns within neuronal populations is also related to the actual detection of visual stimuli and constitutes a second effect in addition to background heterogeneity.

## Temporal consistency of the population code

So far, we have mainly addressed static differences in population activity structure correlating with behavioral responses. However, population codes can show complex temporal properties, such as transient formation of assemblies (*Miller et al., 2014*; *Harris and Mrsic-Flogel, 2013*). After confirming the stability of our recordings to avoid potential confounds (*Figure 1—figure supplement 1*), we addressed whether such temporal population structures might offer additional insight in neural mechanisms of visual detection. We again split the data into miss, fast, and slow response trials, and computed the correlations between response patterns from different trials separately for preferred and nonpreferred neuronal populations (*Figure 6a,b*). Note that this analysis is not sensitive to potential nonstationary effects that might create artificial differences because all stimulus types and behavioral responses are intermingled in time.

Within the preferred population, as well as within the nonpreferred population, we found that neuronal population activity patterns were more similar during fast trials, with a trend for slow trials, than during miss trials (preferred population, miss-slow; p=0.081, miss-fast; p<0.05, nonpreferred population, miss-slow; p<0.05, miss-fast; p<0.05) (*Figure 6c,d*). To rule out that this effect might arise from biases in the analysis, we also compared these population pattern consistencies to those obtained from a shuffling procedure within stimulus types (see 'Materials and methods'). Similarly,

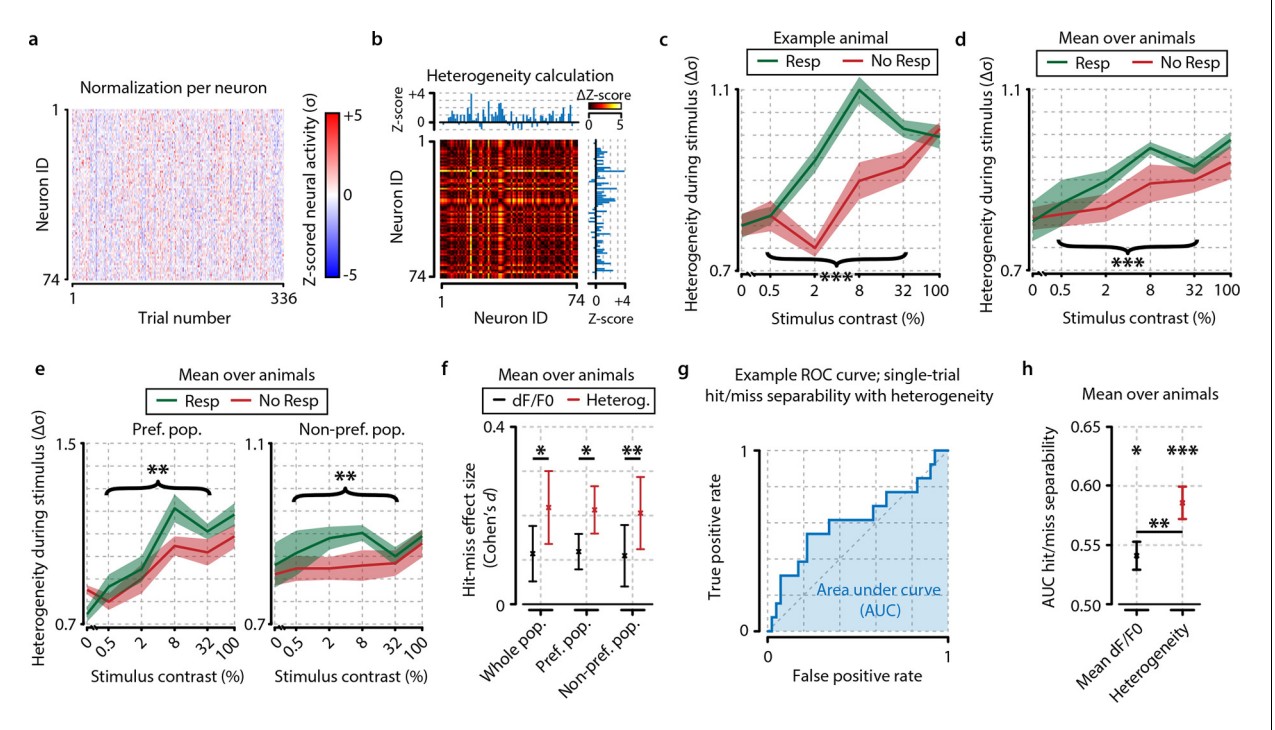

**Figure 3.** Neuronal response heterogeneity within populations correlates better with visual detection than mean preferred population (pref. pop.) activity. (a) Neuronal activity as in *Figure 2d*, but presented as z-score normalized per contrast to be able to compare relative changes across contrasts. (b) Schematic representation of the method to compute heterogeneity on an example trial (see also 'Materials and methods'). The $dF/F_0$ response of each neuron is z-scored per contrast and the distance (absolute difference) in z-scored activity between all pairs of neurons is calculated for each trial (color-coded $\Delta Z$-score). The population heterogeneity in a given trial is defined as the mean $\Delta Z$-score over all neuronal pairs. (c) Population activity heterogeneity in an example animal shows a strong correlation with visual detection. Comparison between detected (resp.) and undetected (no resp.) trials for test contrasts as a group (paired t-test, p<0.001) was highly significant. (d) As (c), but showing mean over all animals (n=8). Stimulus detection correlated with higher heterogeneity; test contrast group hit–miss comparison was highly significant (p<0.001). (e) As (d), but for heterogeneity within the preferred (left panel) and within the nonpreferred (right panel) population only. Hit–miss differences were found in the preferred population (test contrast group, p<0.01) and nonpreferred population (test contrast group, p<0.01) similar to the whole population (d). (f) Using a measure of effect size analysis (Cohen's d), heterogeneity was found to show a stronger correlation with stimulus detection than mean $dF/F_0$ within the whole population (Cohen's $d$=0.114 vs. d=0.218); within the preferred population (Cohen's $d$=0.119 vs. d=0.213) and within the nonpreferred population (Cohen's $d$=0.110 vs. d=0.206) [paired t-tests over animals (n=8) whole population; p<0.05, preferred population; p<0.05, nonpreferred population; p<0.01]. (g) Example receiver operating characteristic (ROC) curve showing the linear separability of single-trial hit and miss trials using population heterogeneity (see 'Materials and methods'). The separability can be quantified by the area under the curve (AUC; blue shaded area). True positive rate: fraction of hit trials classified as hit. False positive rate: fraction of miss trials classified as hit. (h) Statistical quantification of hit/miss separability using either mean $dF/F_0$ (black) or heterogeneity (red) across animals (n=8). Both measures predict the animal's response above chance (FDR-corrected paired t-test $dF/F_0$ and heterogeneity AUC vs. 0.5, p<0.05 and p<0.001, respectively) but behavior can be predicted better using heterogeneity (paired t-test, $dF/F_0$ vs. heterogeneity AUC, p<0.01). All panels: shaded areas/error bars show the standard error of the mean. Statistical significance: *p<0.05; **p<0.01; ***p<0.001.

The following figure supplements are available for figure 3:

**Figure supplement 1.** Contrast-dependent responses of the preferred, nonpreferred, and whole population show distributed hit-correlated modulations in heterogeneity, but modulations in mean $dF/F_0$ are smaller and only significant for the preferred population.

**Figure supplement 2.** Analysis of hit/miss effect size (Cohen's d) shows that simple average perform worst at separating hit from miss responses.

population pattern correlations were significantly higher than shuffled for fast and slow trials, while miss trial consistency was not statistically different from the shuffled control (both preferred and non-preferred populations, paired t-tests shuffled vs. real, miss; p>0.3, slow and fast; p<0.05). However, note that these pattern consistencies are relatively low: they cannot fully account for the population

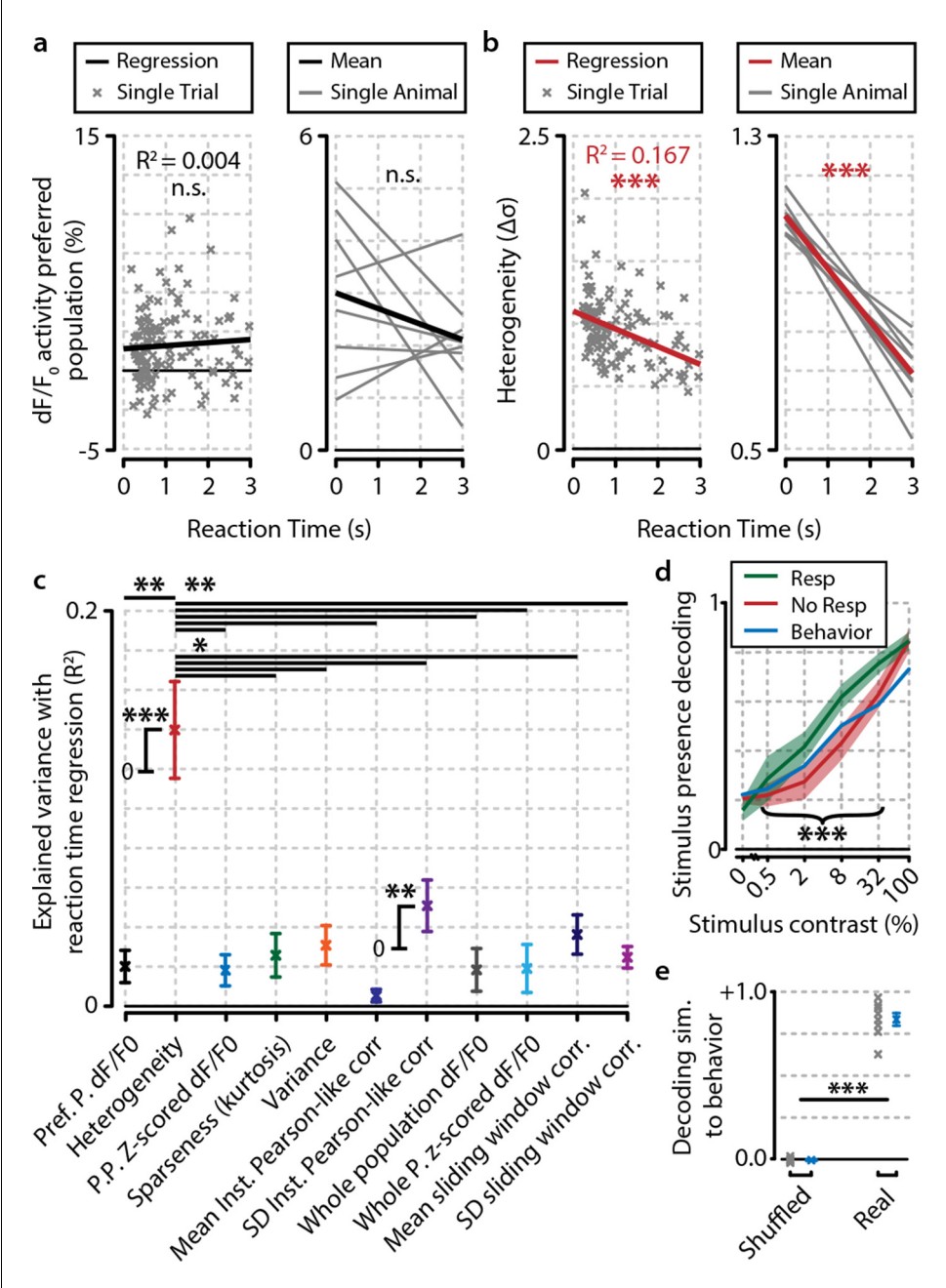

**Figure 4.** Heterogeneity is correlated with reaction time. (a) Reaction time shows no correlation with mean preferred population dF/F₀ activity during stimulus presentation for any individual animal (left panel; example animal) nor for the meta-analysis over animals (right panel; FDR-corrected one sample t-test over individual regression slopes per animal, n=8, n.s.). (b) Same as (a) but for heterogeneity. Heterogeneity shows a strong correlation with reaction time (left panel; example animal, p<0.001) as well as well for the meta-analysis over animals (p<0.001). (a, b) Note different y-axis scaling per panel for display purposes. (c) Comparison of the explained variance of several neural metrics. Only heterogeneity (FDR-corrected one sample t-test, p<0.001) and spread (SD) in instantaneous Pearson-like correlation (see 'Materials and methods') (p<0.01) correlate significantly with reaction time; all other metrics do not [preferred-population (Pref. P.) dF/F₀, preferred-population (P.P.) z-scored dF/F₀, variance, sparseness (kurtosis) mean instantaneous Pearson-like correlation, whole-population (z-scored) dF/F₀, mean and SD of sliding window correlation (width 1.0 s); all n.s.]. Heterogeneity explains more reaction-time-dependent variance than any other metric (FDR-corrected paired t-tests, all p<0.05). (d) Decoding of stimulus presence shows similar accuracy as actual behavioral performance by the animals (*Figure 1e*). When the animal has detected the stimulus (resp.; green line), the decoder is better able to correctly judge its presence (a value of 1 indicates perfect performance, paired t-test, p<0.001). Shaded areas show the standard error of the mean. (e) Behavioral detection performance is more similar (sim.) to the optimal decoder's performance than expected by chance (paired t-test, n=8 animals, shuffled vs. real similarity, p<0.001). Gray: single animal; blue: mean across animals. All panels: error bars/shaded regions show standard error of the mean. Statistical significance: *p<0.05; ***p<0.01; ***p<0.001.

*Figure 4 continued on next page*

*Figure 4 continued*

The following figure supplements are available for figure 4:

**Figure supplement 1.** Of several tested neural metrics, only heterogeneity and spread (SD) in instantaneous Pearson's correlations show a significant relationship with behavioral reaction time (RT).

**Figure supplement 2.** Fidelity of stimulus feature representation and population heterogeneity are correlated with accurate visual detection and are not influenced by neuropil contamination.

activity structure and must therefore be interpreted as happening against a background of dynamic population activity.

## Analysis of heterogeneity in multidimensional space

Most of our results so far have focused on the experimentally observed differences in hit-miss and reaction-time-dependent effect sizes between heterogeneity and mean population activity, but have not addressed the question how this metric might be interpreted within a theoretical framework. Although heterogeneity correlates better with behavioral responses, and especially with reaction time, than many other metrics, this does not exclude that heterogeneity might be an epiphenomenon. We will address this issue next (see also 'Materials and methods', section 'Analysis of multidimensional inter-trial distance in neural activity') using an alternative definition of heterogeneity extended to multidimensional space. This alternative definition is required to study multidimensional properties of population responses, but yields a very similar correlation with stimulus detection as our pairwise definition of heterogeneity (*Figure 3—figure supplement 2*).

Neuronal population activity during any time epoch can be visualized as a single point in multidimensional neural space. For instance, the mean output in spikes per second of a 'population' of two neurons will always be somewhere within a two-dimensional space bounded by the minimal and maximal neural activity of these two neurons (*Figure 7a*). Within a normalized version of this space, a change in mean neural activity will always be parallel to the main diagonal that crosses the origin (minimal neural activity of both neurons) and the point of maximum activity for both neurons. This is true for populations consisting of two neurons, but can be readily extended to any number of dimensions (*Figure 7d*). Heterogeneity, on the other hand, does not change when a point moves across this diagonal (the difference will be zero regardless of whether all neurons are firing at 0 spikes per second, or their maximum), but rather changes as a point moves orthogonally to this diagonal (*Figure 7c*).

The observation we present in our current study—viz. that heterogeneity correlates with hit responses—can be explained by two mutually exclusive hypotheses: (1) the basis for hit and miss-related responses in V1 resides in specific regions in multidimensional neural response space (i.e. discrete states in the neural circuit) and therefore heterogeneity is an epiphenomenon (*Figure 7a,b*), or (2) neuronal response heterogeneity per se is important for stimulus detection. The latter implies that neuronal population response patterns during hit and miss trials should be distributed symmetrically around the main diagonal (which is the gradient along which the mean changes, as well as the axis where heterogeneity is zero). This is because regardless of the specific location in multidimensional space, heterogeneity only captures the distance to this diagonal; rotation or mirroring around this diagonal, therefore, does not change heterogeneity, but does in fact change the distribution in multidimensional space (*Figure 7c,d*).

Accurately quantifying a distribution's shape in multidimensional space requires exponentially more data points as the number of dimensions increases. For direct quantification, our current data set is unfortunately insufficiently large. Therefore, in order to estimate multidimensional symmetry around the main diagonal, we decided to study the effect of mirroring points across this diagonal. First, we calculated the distribution of pairwise inter-point distances in neural response space without mirroring (*Figure 7e*). In this case, each point again represents the population response during a single trial. The data show that the inter-point distance was slightly, but significantly larger for hit than miss trials (paired t-test, $p < 0.05$) (*Figure 7f*). This indicates that population responses during

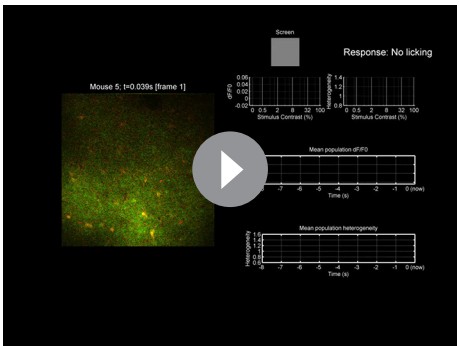

**Video 1.** Typical raw data example showing the first 10,000 recorded frames from animal 5. The left-hand side shows xy-corrected, but otherwise unaltered, raw fluorescence data. The legend above this raw data video shows the time after start of the experiment in seconds and the number of acquired frames. The top two panels on the right-hand side show (left) a depiction of the stimulation screen (gray isoluminant background or oriented drifting grating during stimulus presentation), and (right) whether the mouse is making a licking response. The two panels below show a live updated summary of mean $dF/F_0$ (left) and heterogeneity (right) during each trial. Green indicates a licking response, and red indicates no response. The two lower panels show a live trace of mean population $dF/F_0$ and heterogeneity. Licking responses are shown as red dotted lines, and stimulus presentations are shown as a gray shaded area. Note that the recording is very stable, except during periods of heavy licking, such as after hit responses, when reward is delivered. Also note that neural data acquired during licking are not used for any of our analyses and do, therefore, not influence our results. The mouse is licking vigorously during the initial period of the recording, but more typical behavior sets in less than 2 min after start of the recording. Near the end of the video, it can be seen that hits and misses are more easily separable using heterogeneity than $dF/F_0$ (although this difference is stronger in the example video than in the entire data set as a whole).

hits encompass a larger volume of neural response space than during misses, which will increase heterogeneity values and allows more information to be encoded with the same number of neurons.

To assess symmetry specifically, we mirrored each trial one at a time across the main diagonal and recomputed in each case the distribution of pairwise inter-point distances. If the population responses are distributed asymmetrically around the diagonal, then mirroring will increase the pairwise distance, while if they are distributed symmetrically no change should be observed. There was a small but significant increase in inter-point distance for both hits and misses, and mirroring increased the inter-point distance more for hits than misses (p<0.05; *Figure 7f*, inset). Although the effect sizes are small, they are significant, and we can therefore—at least based on this analysis—not (yet) conclude that heterogeneity is more than an epiphenomenon. In fact, the difference between hits and misses suggests that population responses during hit trials are more asymmetrical (i.e. more clustered in discrete states of neural activity) than during miss trials (*Figure 7f*, inset). Considering that inter-trial population pattern consistencies were lower during miss trials (*Figure 6*), we can conclude that neuronal populations during miss trials show more random behavior within a limited neural space, while neuronal populations during hit trials show more structured behavior in a more extended neural space.

Noting the small effect size of the previous analysis, we asked whether the removal of the mean of the neuronal population response was as detrimental to encoding hit/miss differences as the removal of heterogeneity. Again visualizing population responses as points in neural space, one can remove any differences in mean response between trials by projecting all population responses to a plane orthogonal to the diagonal or remove any differences in heterogeneity by projecting all points onto a manifold at a fixed distance from the diagonal (see 'Materials and methods' and *Figure 7g*). To test the effect of these removals on hit/miss differences, we performed a decoding procedure of hit versus miss trials (i.e. we decoded the animal's response) on the original data, and on data with the mean, the heterogeneity or both aspects removed. The results show no difference between the original data and the mean-removed data, but removing heterogeneity (or both heterogeneity and the mean) led to a small but significant decrease in decoding performance (heterogeneity-removed vs. original data, p<0.05; heterogeneity-removed vs. mean-removed, p<0.05) (*Figure 7h*). However,

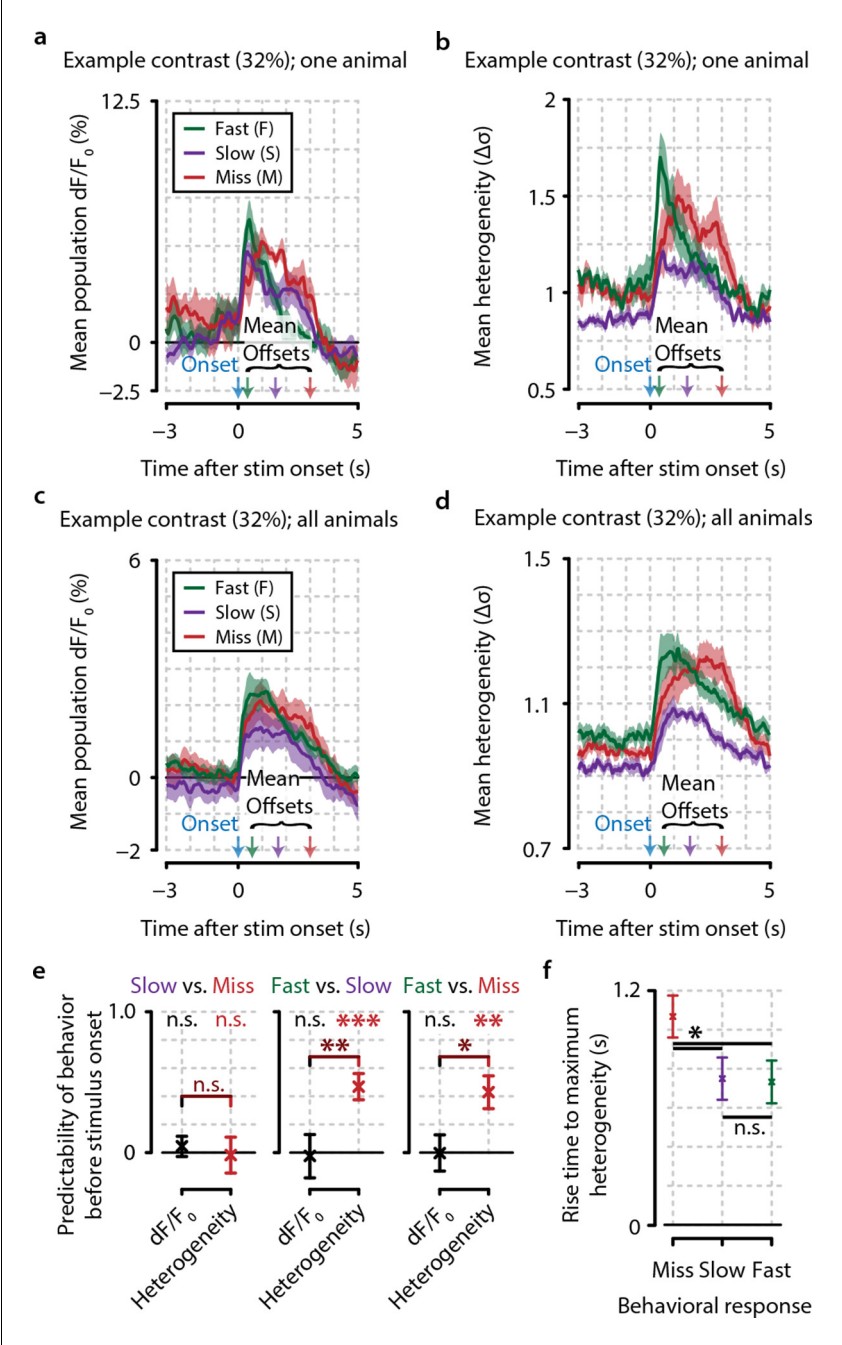

**Figure 5.** Heterogeneity preceding stimulus onset predicts behavioral reaction time. (**a, b**) Example traces from one example animal for population dF/$F_0$ (**a**) and heterogeneity (**b**) of fast (F, green), slow (S, purple), and miss (M, red) responses (mean ± standard error over trials) also showing mean stimulus offsets. (**c, d**) As (**a, b**) but showing mean ± standard error over animals. (**e**) Fast behavioral responses are predictable before stimulus onset using heterogeneity [FDR-corrected one-sample t-tests vs. chance level (0); S–M, p=0.799; F–S, p<0.001; F–M, p<0.01] but not using dF/$F_0$ (FDR-corrected one-sample t-tests vs. chance level; S–M, p=0.157; F–S, p=0.811; F–M, p=0.924; FDR-corrected paired t-test for heterogeneity vs. dF/$F_0$; S–M, p=0.477; S–F, p<0.01; F–M, p<0.05). (**f**) The population heterogeneity during stimulus presentation does not merely reflect a continuation of pre-stimulus neural state; detected stimuli (slow and fast responses) elicit a faster rise to the maximum heterogeneity level than undetected stimuli (miss trials) (paired t-tests, n=8 animals, p<0.05). Slow and fast responses do not differ significantly (p>0.05). All panels: error bars/shaded areas indicate standard error of the mean. Statistical significance: *p<0.05; **p<0.01; ***p<0.001.

The following figure supplement is available for figure 5:

**Figure supplement 1.** Behavioral response is predictable on single trials when using heterogeneity, but not when using dF/$F_0$.

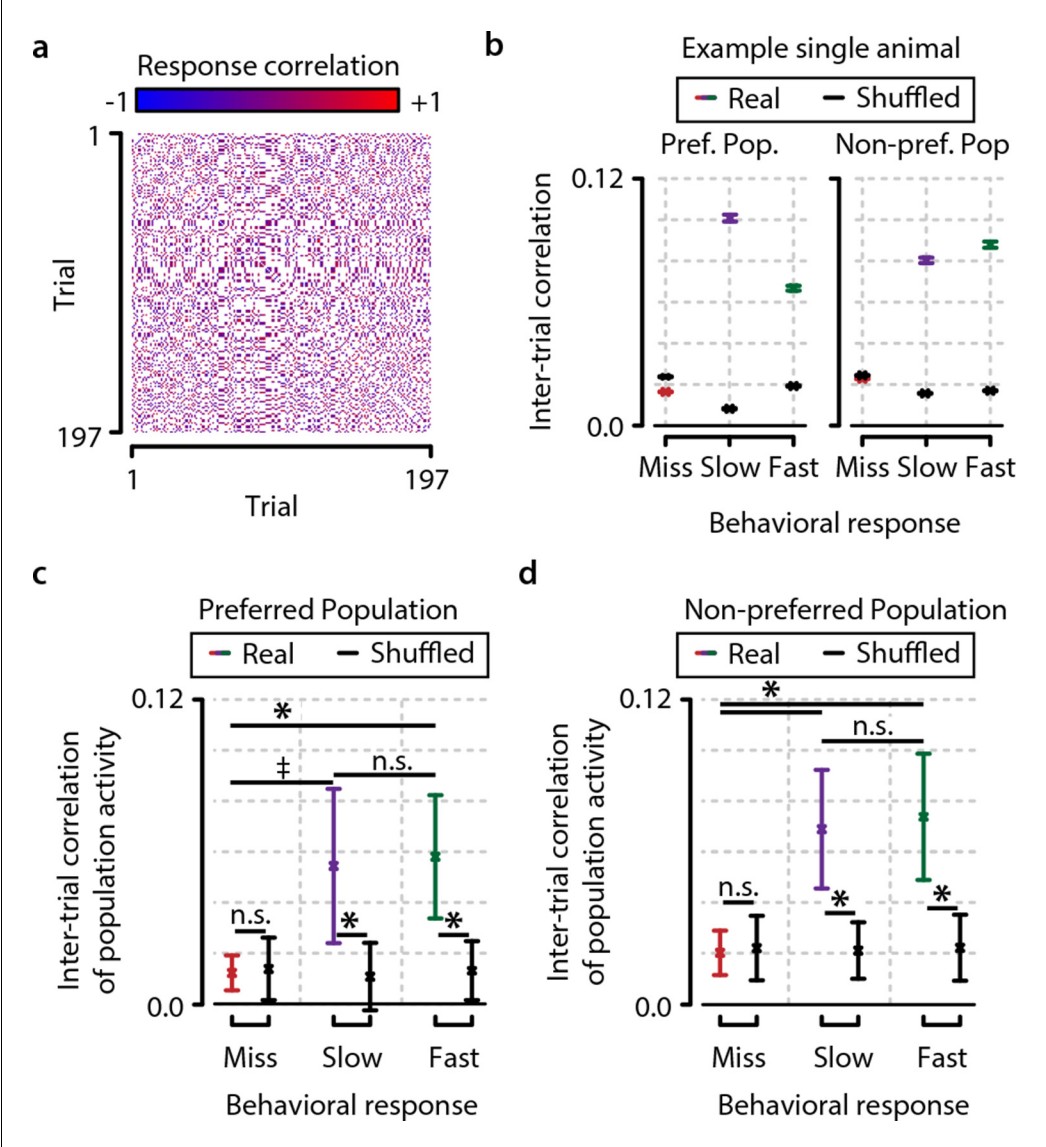

**Figure 6.** Consistency in population activation patterns across trials is increased during hit trials (fast and slow) compared to miss trials. (**a**) Data from example animal showing inter-trial correlations (Pearson's r) between population responses to same-orientation stimuli (pooled over test contrasts only). (**b**) Data from same animal as in panel (**a**), showing population higher activity pattern consistency (mean ± standard error over trial pairs) for fast and slow response trials than miss trials within both the preferred and nonpreferred population. Colored lines show real data, and black lines show shuffled data (see text and 'Materials and methods'). (**c, d**) Inter-trial correlations (mean ± standard error over all animals) as quantification of population activation pattern consistency are significantly higher during fast trials (with a trend for slow trials) than during miss trials within the preferred as well as the nonpreferred neuronal population, suggesting that visual stimulus detection is correlated with the occurrence of more stereotyped population responses (FDR-corrected paired t-tests, preferred population; miss-slow, p=0.081; miss-fast, p<0.05; slow-fast, n.s.; nonpreferred population; miss-slow, p<0.05; miss-fast, p<0.05; slow-fast, n.s.). Comparison with correlations of shuffled data yielded similar results (both preferred and nonpreferred populations; paired t-test real vs. shuffled: miss; p>0.2, slow and fast; p<0.05). Error bars indicate standard error. Statistical significance: * p<0.05; ‡ 0.05<p<0.1.

even with heterogeneity removed, decoding performance was still well above chance (63% correct for original data, 60% correct for both removed; 50% is chance).

We, therefore, conclude that heterogeneity contributes significantly as a non-epiphenomal population property to the differentiation in neural responses between visual detections and failures to detect a stimulus, but also that most information resides in neuronal population response patterns other than its mean or heterogeneity. Moreover, the mean response of a neuronal population is less

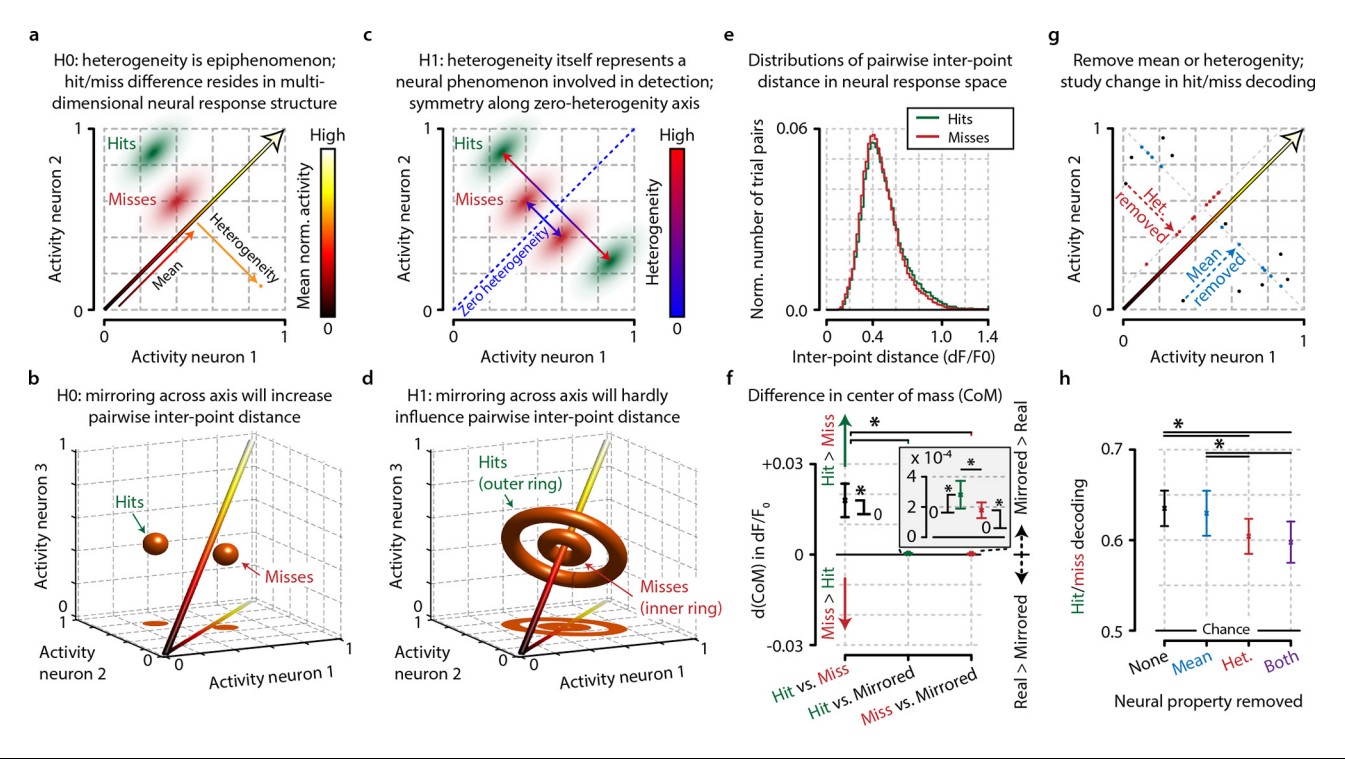

**Figure 7.** Conceptual interpretation of heterogeneity as neuronal population coding phenomenon. (a–d) The mean population activity during a certain time epoch can be visualized a single point in multidimensional neural space, where every axis represents the activity of a single neuron. (a) For an example population of two neurons, the main diagonal (arrow) represents the line along which the mean population activity changes. Orthogonal to this line is the gradient along which heterogeneity changes, representing the distance of each point to the main diagonal. The effects of heterogeneity on hit/miss differentiation as reported in this study could be epiphenomenal if the real underlying differentiation depends on localized, segregated clusters of neural activity for hits (green cloud) and misses (red cloud). (b) This principle can be extended to multidimensional space; segregated clusters activity will show asymmetrical distributions of population activity around the diagonal. (c, d) Alternatively, heterogeneity itself could represent a fundamental characteristic of hit/miss differences; in this case, population responses should be distributed symmetrically around the diagonal (see text for more explanation). (e) Calculating the pairwise inter-point distance (each point being the population activity during a single trial) can reveal information about the underlying multidimensional structure of neuronal population activity. Green: distribution of inter-point distances for hit trials; red: same for miss trials. (f) Population responses during hit trials are distributed within a larger volume of neural space, as shown by the on average larger inter-point distance for hits than misses [paired t-test, difference in center of mass (d(CoM) hit vs. miss, p<0.05]. Mirroring point across the diagonal to assess symmetry shows a small, but significant asymmetry for hits and miss (both p<0.05) and larger asymmetry for hits than misses (p<0.05). This suggests that neuronal populations during hit trials show more structured behavior in a more extended neural space than during miss trials. (g) Schematic representation of how the mean, heterogeneity, or both can be removed from population responses to assess the effect they have on hit/miss separability (see also text and 'Materials and methods'). (h) Removing heterogeneity impairs hit/miss decoding more than removing the mean (paired t-test, p<0.05) but in all cases (including removing both) the hit/miss separability is still well above chance (0.5). This suggests that heterogeneity is more important than population mean activity for differentiating stimulus detection from non-detection, but that other more complex neural phenomena account for most of the population response structure. All panels: error bars indicate standard error. Statistical significance: *p<0.05.

important than its heterogeneity. While neural response heterogeneity may be an important factor and useful metric for its strong correlation with especially reaction times, further research is required to discover which other neural properties may be important for visual stimulus detection.

## Discussion

We found that behavioral stimulus detection correlates more with nonlinear neuronal population activation patterns, such as heterogeneity, correlations, and variance, rather than overall response strength in L2/3 of mouse V1 (*Figures 2* and *3*; *Figure 2—figure supplement 1*, *Figure 3—figure supplement 1*, *Figure 3—figure supplement 2*). Using a novel measure of population heterogeneity, we show that the differentiation in activation within these populations predicts visual detection,

and particularly behavioral reaction time, and is associated with an increased accuracy of stimulus presence and feature representation by the population (*Figure 4*; *Figure 4—figure supplement 2*). High heterogeneity prior to stimuli correlated with fast hit responses, but also showed a dissociation between detection and nondetection behavior, indicating that detection-related population activity may be gated by arousal mechanisms (*Figure 5*; *Figure 5—figure supplement 1*). Neuronal population activation patterns are more similar during accurate task performance upon repetition of the same stimulus, but not when the animal fails to respond, suggesting that specific population patterns may recur when the animal is well engaged in the task (*Figure 6*). Taken together, these results suggest that neural processing of information related to detection behavior depends on transient differentiation in neuronal activity within cortical populations rather than on temporally stable ensembles or on gain modulation of population activity as a whole (*Figure 7*).

## Potential confounds

Our analyses show differences between hit and miss trials that we interpret as being related to perception and visual processing. However, in principle, the observed differences could be subject to a number of confounds that might limit their interpretability. First, the relatively mild water restriction led to a behavioral performance at the 100% probe trials that is lower compared to other studies using similar tasks (*Glickfeld et al., 2013*). This could mean that the observed differences in heterogeneity between hit and miss trials are due to changes in motivation rather than visual detection. If this were the case, however, hit–miss differences should be as large during 100% probe trials as during test contrast trials. *Figure 3* and *Figure 3—figure supplement 1f,2* shows that this is not the case; during intermediate test contrasts (2% and 8%), the hit–miss difference in heterogeneity is largest. A similar reasoning applies to 0% contrast probe trials, where a heterogeneity difference between (false alarm) responses and correct rejections was lacking, which strongly argues against heterogeneity being due to response emissions per se. Thus, it is highly unlikely that the neural correlates of behavioral responses as reported in this study are due to differences in motivation. Given this caveat on suboptimal performance, one may predict that a better behavioral performance would have only increased the hit–miss effect sizes we report in this study.

Other potential confounds include instabilities in z-plane focus, other locomotor-related artifacts and running-induced modulations, as it has been reported that behavioral activity and running can induce instabilities in the plane of focus with awake two-photon calcium recordings, as well as changes in the neuronal responses in mouse V1 (*Dombeck et al., 2007*; *Niell and Stryker, 2010*; *Saleem et al., 2013*). To address potential z-shifts during the acquisition of neural data, we compared each imaging frame to 3D anatomical z-stacks acquired after recording the neural data (see 'Materials and methods'). Slight changes in z-location were detected after the onset of hit responses by licking, but these were mostly confined to the reward presentation period (which was not used for any of our analyses) and rarely exceeded more than a couple of microns (*Figure 1—figure supplement 1*). Moreover, exclusion of trials where mice were running did not qualitatively change hit–miss differences in neuronal activity (*Figure 2—figure supplement 1g,h*), nor did using only the first 400 ms after stimulus onset to avoid licking-preparation feedback from other brain regions (*Figure 2—figure supplement 1i, j).*

To control for potential confounds related to eye movements and blinking, we analyzed eye-tracking videos to detect blinks and saccades. After removing all trials where the animals were making saccades or were blinking, we again found no qualitative difference with the results we observed previously (*Figure 2—figure supplement 1k,l*), but did observe a small but significant correlation between pupil size and heterogeneity, suggesting higher heterogeneity with increased arousal states (*Figure 4—figure supplement 1e*). Overall, we conclude that the neural correlates we report here, and interpret as related to perception, are most likely not due to recording instability, changes in motivation, locomotion, motor-related signals associated with licking, or eye movements and blinking.

## Possible cortical layer specificity of poor correlation of mean population responses

Importantly, our results only pertain to L2/3 of the primary visual cortex in mouse, which does not exclude the possibility that the mean population response of, for example, deeper layers (L5) in V1

would correlate better with visual stimulus detection. Previous research has shown that extensive differences exist between superficial and deep layers: whereas L2/3 neurons often show relatively low peak firing rates and sparse responses to sensory stimulation, L5 neurons show denser response patterns with on average higher peak firing rates (*De Kock and Sakmann, 2008*; *Harris and Mrsic-Flogel, 2013*). In somatosensory cortex, it has been shown that hits and correct rejections in a go-no-go object localization task can be better separated using mean spiking rates in L5 than in L2/3 (*O'Connor et al., 2010*). Our result that L2/3 populations show only a small differentiation between hit and miss responses in mean activation should therefore not be taken as proof for a canonical principle also applicable to other cortical layers. Future validation of our results in deep layers is necessary for a decisive answer whether our results are indeed applicable to different layers of primary sensory cortex.

## Comparison with other studies and neural interpretation of heterogeneity

Our task design included drifting gratings with different orientations, with the qualification that orientation was task-irrelevant for the mice, as they were only required to detect stimuli whenever they appeared. Our observation that stimulus features are represented more accurately (as quantified by decoding accuracy) during hit than miss trials may therefore be somewhat surprising (*Figure 4—figure supplement 2a*). This suggests that mechanisms that increase the likelihood of stimulus detection may be acting through a general enhancement of stimulus processing intensity, corroborating previous research in monkey showing that attention can lead to horizontal shifts in contrast response curves, as if the stimulus were of higher contrast (*Martínez-Trujillo and Treue, 2002*). It is interesting to ask whether our results on heterogeneity can be cast in terms of dynamic range effects. Neurons are expected to climb in this dynamic range when visual contrast increases, which is confirmed by the rise in dF/F$_0$ (*Figure 2d*). However, if heterogeneity would be primarily determined by neurons being able to operate along the steep slope of their dynamic range, then the large difference in heterogeneity between hits and misses (*Figure 3c*) along the test contrasts (0.5–32%) would not be expected.

Of further interest is to compare our results on heterogeneity with studies reporting that sparseness in L2/3 populations of rodent V1 is high during passive viewing (*Barth and Poulet, 2012*) depends on cortical state and improves neural discriminability during passive processing of natural scenes (*Froudarakis et al., 2014*). Although in our analysis sparseness and variance explained more behavioral variability in reaction time than (z-scored) mean population activity (*Figure 4c*), these measures perform much worse than heterogeneity and the spread of instantaneous Pearson-like correlations. Possibly, a sample of ~60–70 tuned neurons is insufficient to estimate instantaneous sparseness accurately. An alternative explanation for this poor correlation could be that sparseness of L2/3 populations results from anatomical wiring required for efficient stimulus coding and to enable locally selective synaptic plasticity without immediately changing the coding of stimulus features within the population response (*Rao and Ballard, 1999*). Correlates of visual detection that depend on accurate stimulus feature representation might then be better captured by a maximization of Mahalanobis distances in neural activity between pairs of neurons within this already sparse network. This latter interpretation is in line with the data we recorded and suggests that sparse stimulus representation by L2/3 neurons reflects a structural optimization of the population code to represent stimulus features, while heterogeneity captures more temporally dynamic modulations related to perception.

## Multidimensional analysis: heterogeneity contributes to but does not fully account for visual detection

In addition to our approach based on pairwise relations in neuronal responses, we investigated multidimensional patterns of population activity (*Figure 7*). These results indicate that, while heterogeneity is more important for separating stimulus detections from nondetection in neural response space than the population mean, these properties combined still cannot capture the full set of neuronal response characteristics that define the accurate detection of visual stimuli in L2/3 of mouse V1. This suggests that other patterns of population activity, such as potentially transient assembly formation, may be important for visual stimuli to be correctly detected. From our multidimensional

analyses, we can conclude that simple bulk approaches (i.e. correlating a population's mean response with behavioral output) are insufficient when one aims to address how early sensory cortex areas are involved in the processing and detection of visual stimuli.

Related to these findings is our observation of behavioral-state-specific consistencies in population activation patterns across trials. This provides some constraints on how population heterogeneity is modulated at a neurophysiological level. Neuromodulators such as acetylcholine (ACh) and noradrenaline are correlated with attention and arousal, and may influence cortical population dynamics (*Metherate et al., 1992*; *Coull et al., 2004*; *Pinto et al., 2013*), such that they facilitate repeated activation of similar subnetworks of neurons within a population responding to the same stimulus. Without such neuromodulators, neurons within the same preferred population would randomly participate in representing the current stimulus. This interpretation is compatible with previous work; for instance, ACh has been observed to influence burst spiking, membrane potential fluctuations, cortical oscillations, and desynchronization. These processes have been implicated in modulating competitive inhibition effects within neuronal populations and may very well influence the consistency of specific neuronal subnetworks being activated (*Borgers et al., 2008*; *Fries, 2009*; *Bosman et al., 2014*). If heterogeneity in a recurrently connected V1 population is in part determined by suppression of the most weakly by the most strongly stimulus-driven neurons, then behaviorally correlated heterogeneity enhancements may be another facet of arousal as well as perception-related modulations of stimulus-evoked population activity.

Population coding phenomena have long been hypothesized to be important for sensory processing, but so far few studies have investigated their relevance for perceptual decisions. Here, we show that population heterogeneity is correlated with behavioral stimulus detection and that it predicts correct behavioral performance. Our results imply that neurophysiological measures dependent on population averages (i.e. multiunit activity, EEG, and fMRI) may underestimate the correlation between visual detection and V1 L2/3 activity because the assumption of population response homogeneity is violated especially during active processing of visual information. In short, our results support contrast-sensitive changes in mean population activity during visual task performance (*Figure 3c,d*), but stress the importance of population recordings with single-cell resolution (*Figure 4c–f*).

## Materials and methods

### Animals and surgery

All experimental procedures were conducted with approval of the animal ethics committee of the University of Amsterdam (cf. *Goltstein et al., 2013*; *Montijn et al., 2014*). Experiments were performed on eight adult, male wild-type C57BL/6 mice (Harlan), 128–164 days old at the day of calcium imaging (29.1–32.7 g). Prior to the imaging experiment, all animals were surgically fitted with a head-bar implant and trained head-fixed for up to 3 months to perform a visual go/no-go detection task. At the day of the imaging experiment, we performed intrinsic signal imaging to define the area corresponding to the retinotopic region in V1 responsive to the visual stimulus. We performed a small (1.5–2.0 mm) craniotomy at that location and used multicell bolus loading with Oregon Green BAPTA-1 AM to record calcium transients and sulforhodamine-101 (SR101) to label astrocytes (*Stosiek et al., 2003*; *Nimmerjahn et al., 2004*).

### Behavioral training

Mice were trained 5 days per week, each for approximately 45 min per day, on a head-fixed go/no-go visual detection task over a period of 10–12 weeks, where we aimed to get sufficient hit as well as miss trials for test contrasts. Mice were water-deprived for 6 h preceding training and otherwise had ad libitum access to water. Weight was monitored three times per week and never dropped below 90% of their nonrestricted growth curve. Behavioral training was performed inside four dark, sound attenuated chambers and occurred during the active (dark) cycle of the animals; each animal was always trained in the same behavioral setup. We did not observe any deviant learning effects associated with any specific behavioral setup (data not shown). During the first five days of training, we conditioned licking in response to visual stimulation by pairing passive stimulation with reward delivery (~9 μl of water with 15% sucrose with 1% vanilla extract) (stage 1). After the conditioning

phase, visual stimuli (100% contrast drifting gratings as described in the previous paragraph) were presented indefinitely until mice made a licking response that was monitored using a custom-built infrared LED-based lick detector. When animals made a response, the visual stimulus presentation terminated and reward was available for 5 s. This shaping phase (stage 2) lasted for a maximum of 5 days or less if the animals were often making clear lick responses. After this ~2-week initial phase, we started training the animals on a simple version of the final task (stage 3); maximum stimulus presentation was reduced to 5 s and subsequent trials would only start if the mice did not make any lick responses for at least a random interval of 1–3 s. During this stage, reward size was gradually reduced to ~3 µl per trial. When animals would consistently perform at least 80 trials within a period of 45 min, they would be moved to the next stage. In stage 4, we introduced 0% contrast probe trials to monitor the behavioral performance of animals by testing for false-alarm responses and calculating if they showed statistically significant above-chance performance. In this stage, we also lengthened the inter-trial interval to any random duration between 6 and 8 s. Once mice attained a sufficient ratio of hit/miss trials, we moved them to training stage 5, where we increased the inter-trial interval to 10–12 seconds and presented mild air puffs as a negative reinforcer whenever mice would lick outside the stimulus presentation or reward delivery period. At this stage, animals were required to not lick for a random interval of 1–3 s in order to gain access to the next stimulus presentation. Stage 5 lasted until the mice had been trained for 8–10 weeks in total. Finally, if mice performed consistently and significantly above chance during stage 5 (n = 12 / 21 animals), then in the 2-week period preceding the imaging experiment mice were trained on the microscope setup, and our setup's resonant mirrors were activated to produce the characteristic 8000 Hz sound that would also be present during calcium imaging. In this final stage, all possible efforts were made to simulate surroundings of the eventual calcium imaging experiment as closely as possible to habituate the mice to the two-photon laser lab's environment. Mice were always allowed to take up to 3 s after stimulus onset to respond and were thus not explicitly trained to make fast behavioral responses.

## Craniotomy and dye injection

On the day of the two-photon calcium imaging experiment, buprenorphine (0.05 mg/kg) was injected subcutaneously 30–60 min before induction of anesthesia with isoflurane (4.0% induction, 0.8% maintenance during intrinsic signal imaging, 1.5–2.5% maintenance during invasive surgical procedures). After induction, the animal was placed in a custom-built head-bar holder designed for performing surgical procedures. We removed the cover glass, silicon elastomer, and layer of glue covering the skull in the cranial window before performing intrinsic signal imaging to localize the precise location of our stimulus' receptive field location in the primary visual cortex (V1). We subsequently performed a small (1.5–2 mm) craniotomy above the retinotopic area responding to visual stimulation with drifting gratings. After the craniotomy, the dura was kept wet with an artificial cerebrospinal fluid (ACSF: NaCl 125 mM, KCl 5.0 mM, MgSO$_4$ * 7 H$_2$O 2.0 mM, NaH$_2$PO$_4$ 2.0 mM, CaCl$_2$ * 2 H$_2$O 2.5 mM, glucose 10 mM) buffered with HEPES (10 mM, adjusted to pH 7.4). After making the craniotomy, multicell bolus loading with Oregon Green BAPTA-1 AM (OGB) and SR101 was performed 230-270 µm below the dura as previously described in *Montijn et al., 2014* and *Goltstein et al., 2013*. After injection of the dyes, the exposed dura was covered with agarose (1.5% in ACSF) and sealed with a circular cover glass that was fixed to the skull using cyanoacrylate glue. The animal was allowed to recover for a minimum of 90 min before starting the behavioral task and two-photon calcium imaging. Of the 12 mice that learned the task, 2 animals were rejected due to insufficient imaging quality.

## Visual stimulation

All visual stimulation was performed on a 15 in. TFT screen with a refresh rate of 60 Hz positioned at 16 cm from the mouse's eye, which was controlled by MATLAB using the PsychToolbox extension (*Brainard, 1997*; *Pelli, 1997*). Stimuli consisted of sequences of eight different directions of square-wave drifting gratings that were monocularly presented in randomized order. Visual stimulus duration started at infinite during the initial training phase and was gradually reduced to a maximum duration of 3 s for the final task stage. Stimuli were alternated by a blank inter-trial interval of variable duration (random minimum of 10–12 s) during which an isoluminant gray screen was presented. Visual drifting gratings (diameter 60 retinal degrees, spatial frequency 0.05 cycles/°, temporal

frequency 1 Hz) were presented within a circular cosine-ramped window to avoid edge effects at the border of the circular window. A field-programmable gate array (OpalKelly XEM6001, Opal Kelly Incorporated, Portland, OR) was connected to the microscope and behavioral setup and interfaced with the visual stimulus presentation computer to synchronize the timing of visual stimulation with the microscope frame acquisition and behavioral setups.

## Z-drift quantification and recording stability analysis

Slow z-drifts were quantified by comparing the similarity of 100 frames in the beginning, middle and end of each stimulus repetition set to slices recorded at different cortical depths (step size ~1–2 μm) before or after functional calcium imaging was performed for five of eight animals. If z-drifts larger than 10 μm occurred slowly over multiple repetition blocks, or if slow z-drift was detected manually, the entire recording of a single animal was split into multiple analysis periods (n=2 populations for animals 1 and 7; n=1 population for all other animals) and analyzed independently (*Figure 1—figure supplement 1*). For the two animals for which we split the recordings, we afterwards averaged all measures over the two populations, yielding a single independent data point also for these animals for each measure.

To confirm the stability of our recordings, we performed a further analysis quantifying the discriminability of neurons relative to their surroundings over time (*Figure 1—figure supplement 1*). Therefore, we calculated during each imaging frame the mean fluorescence of the pixels within the neuron's soma ($F_{soma}$) and the fluorescence of a neuropil annulus surrounding the soma ($F_{neuropil}$), which we defined as all pixels within a concentric band from 2–5 μm away from the soma. For each frame, we then calculated the discriminability ratio $D_r$ as $D_r = F_{soma} / (F_{soma} + F_{neuropil})$, and set a threshold at $D_r=0.5$ (equal luminance of soma and neuropil). Whenever this measure dropped below the threshold, we calculated the duration of this epoch until it would return to above the threshold, and took the maximum duration of all these epochs as a single measure per neuron. Most neurons from all sessions showed maximum below-threshold durations near 0 s, and no neurons showed durations longer than 1 s (*Figure 1—figure supplement 1*).

To address the potential confound of fast changes in z-plane due to anticipatory fidgeting behavior by the animals, we calculated the depth of each imaging frame and analyzed whether responses to visual stimuli were preceded by shifts in z-plane that could influence our results. As can be seen in figure supplement 1-1L–O, z-shifts were mostly confined to the epoch immediately following hit responses, which are not used in our analyses, and in general z-shifts were very small and rarely exceeded more than 1 μm.

## Eye tracking

We recorded eye movements during the entirety of the calcium imaging experiment to be able to correct for possible contamination of our results by excessive blinking and/or saccades. For this purpose, we placed a near-infrared light sensitive camera (JAI CV-A50IR-C Monochrome 1/2" IT CCD Camera, JAI A/S, Germany) with a large-aperture narrow-field lens (50 mm EFL, f/2.8) above the visual stimulation screen directed at the mouse's visually stimulated eye. Images were acquired at 25 Hz and pupil tracking was performed offline using custom-written MATLAB scripts. Eye position was used to control for possible saccade effects (*Figure 2—figure supplement 1k,l*), and pupil diameter was used to assess its correlation with heterogeneity (*Figure 4—figure supplement 2e*).

## Calcium imaging recordings and final task parameters

Dual-channel two-photon imaging recordings (filtered at 500–550 nm for OGB and 565–605 nm for SR101; see *Figure 1d*) with a 512 x 512 pixel frame size were performed at a sampling frequency of 25.4 Hz. We used an in vivo two-photon laser scanning microscopy setup (modified Leica SP5 confocal system) with a Spectra-Physics Mai-Tai HP laser set at a wavelength of 810 nm to simultaneously excite OGB and SR101 molecules, as previously described (*Montijn et al., 2014*) in cortical layer 2/3 at depths from the pia mater ranging from 140 to 170 μm (*Figure 1—figure supplement 1*, *Video 1*). During data acquisition, mice were performing a go/no-go stimulus detection task where the animals had to lick whenever a visual stimulus was presented. Stimulus parameters were equal to those described above. We varied the contrast of the drifting grating (0%, 0.5%, 2%, 8%, 32%, and 100%) to elicit a wide range of hit/miss ratios. Responses to 0% contrast probe trials were not

rewarded, but responses to all other contrasts were. We did not explicitly aim for very high detection performance (high hit rates and low miss rates) to avoid overtraining and associated habitual or automated responding (*Balleine and Dickinson, 1998*). A complete set of visual stimuli, therefore, consisted of 48 trials (6 contrasts times 8 directions). The order of presentation of these 48 trials was randomized independently for each repetition block. After the experiment was completed, we tested for statistically significant stimulus detection performance by calculating the binomial 2.5th–97.5th percentile intervals (henceforth 95% CI) of response proportion to the two probe trial types—100% and 0% contrast stimuli—using the CP method. Of the 10 animals from which we recorded calcium imaging data during task performance, one was rejected because of excessive variability in responses due to brain movement and one was rejected due to insufficient discriminability between the two types of probe trials (overlapping CIs). All data we present in this paper are from the remaining eight animals. The number of repetitions per stimulus type (unique orientation x contrast) ranged from 6 to 16. For most analyses, we took the mean over all orientations (n=4), so each contrast was presented 24–64 times. For all analyses of single-animal data, each trial was taken as a single data point, where its value was the mean $dF/F_0$ over all recorded frames during stimulus presentation (which was dependent on the reaction time of the mouse). To avoid the confound of having higher signal-to-noise ratios for miss than hit trials due to longer data acquisition, within each contrast group we randomly assigned to all miss trials a duration randomly selected from the reaction time distribution of hit trials.

## Data preprocessing

After a recording was completed, small x–y drifts were corrected offline with an image registration algorithm (*Guizar-Sicairos et al., 2008*). To retrieve $dF/F_0$ values from the recordings, regions of interest (ROIs; neurons, astrocytes, and blood vessels) were determined semiautomatically using custom-made MATLAB software for each repetition block separately (see https://github.com/JorritMontijn/Preprocessing_Toolbox). For these ROIs, we subsequently calculated $dF/F_0$ values as previously described (*Montijn et al., 2014*): For each image frame $i$, a single $dF_i/F_{0i}$ value was obtained for each neuron by calculating the baseline fluorescence ($F_{0i}$), taken as the mean of the lowest 50% during a 30 s window surrounding image frame $i$. $dF_i$ is defined as the difference between the fluorescence for that neuron in the given frame and the sliding baseline fluorescence ($dF_i = F_i - F_{0i}$) (*Montijn et al., 2014*). The mean number of simultaneously recorded neurons/session was 92.6 [range 68 – 130 (SD: 19.0) neurons]. After this initial analysis, all neurons were tested on consistency for preferred stimulus orientation and any neurons that showed inconsistencies over different repetition blocks (i.e. more than one-third showing different preferred orientations) were rejected from further analysis [mean number of consistently tuned neurons per animal was 66.3 ± 18.6 (70.8% ± 7.75% of all neurons) (mean ± SD)]. Unless otherwise specified, all analyses shown in this paper are based on across-animal meta statistics based on a set of eight independent data points (one data point/animal) and all multiple comparison t-test p-values were adjusted by the Benjamini and Hochberg FDR correction procedure and were deemed significant if the resultant p-value was <0.05. For quantification and control procedures related to z-drift and recording stability, see *Figure 1—figure supplement 1*. For control analyses where we performed neuropil fluorescence subtraction (*Figure 4—figure supplement 2i*), we used similar procedures as described previously (*Greenberg et al., 2008*; *Mittmann et al., 2011*); we calculated the correlation (r) between each neuron's somatic fluorescence and surrounding neuropil (annulus between 2 and 5 µm from soma) and corrected on each frame the neuron's fluorescence as follows: $F_{corr} = F_{soma} - r * F_{neuropil}$. Estimated neuropil contamination varied widely between neurons, but was generally in the range between 0.1 and 0.6, similar to previously reported values (*Greenberg et al., 2008*; *Mittmann et al., 2011*). We recomputed the explained variance of several metrics as a function of reaction time (see *Figure 4c*, *Figure 4—figure supplement 1*) and found that neuropil correction did not affect our main conclusions (*Figure 4—figure supplement 2i*).

## Linear regression analysis

All linear regressions were performed on single-animal data sets, yielding regression coefficients for the intercept and slope through minimizing the error between a linear function and the single animal's data points. Statistical significance was quantified by performing a one-sample t-test of the

coefficients from all animals (n=8). Significance level was set at an α of 0.05 and p-values were adjusted if necessary by a post hoc Bonferroni–Holmes correction.

## Calculation of preferred stimulus orientation

We presented eight directions of visual drifting gratings and calculated the preferred stimulus orientation of all neurons by summing opposite directions as belonging to the same stimulus type because the vast majority of mouse V1 neurons is tuned sharply to an axis of movement, but much less so to a specific direction within that axis (i.e. most neurons are strongly orientation-tuned, but less direction-tuned; e.g. *Andermann et al., 2011*). For these four orientations, we took each neuron's mean response over all trials and defined its preferred orientation as the stimulus that caused the highest mean $dF/F_0$ value. For most analyses, we used the neuronal responses to all orientations, except for *Figure 2c,d*, where we used only the response of the preferred orientation, as we hypothesized the preferred population might yield stronger hit/miss differences in neuronal activity.

## Predictability of hit modulation by consistent neuronal responses and whole-population fluctuations

To investigate the source of hit-related increases in population $dF/F_0$ and determine whether there might exist a subgroup of neurons that consistently enhances its activation during detection trials (as compared to nondetection), we defined a $dF/F_0$ hit modulation index $\Psi$ for each hit trial ($t$) for each neuron ($i$) as the neuron's $dF/F_0$ activity ($R$) relative to the mean ($\mu$) and standard deviation ($\sigma$) of its response during miss trials ($m$) of the same type [identical orientation ($\theta$) and contrast ($c$)]:

$$\Psi_{i,t} = (R_{i,t} - \mu_{m,c,\theta}) / \sigma_{m,c,\theta} \qquad (1)$$

In other words, $\Psi_{i,t}$ of a given trial represents the z-scored $dF/F_0$ activity relative to the neuron's response to the same stimulus when the stimulus remained undetected (*Figure 2e*, left panel). The hit-modulation matrix $\Psi$ of all hit trials and all neurons can then be approximated by neuron identity (mean over trials), trial-by-trial fluctuations (mean over neurons), or both (addition of the matrices yielded by the two previous approximations) (*Figure 2e*). We then calculated the explained variance ($R^2$) of the population response pattern by its canonical equation based on the residual ($SS_{res}$) and total sum of squares ($SS_{tot}$). We defined $SS_{tot}$ as the sum of all squared values in $\Psi$, and $SS_{res}$ as the sum of the squared differences between $\Psi$ and the approximation matrix as defined above (by neurons, trials, or both). To assess significance, we performed 1000 shuffle iterations where we randomized neuronal identities per trial (for approximation by neuron identity), randomized trial identities per neuron (for approximation by trial identity), or randomized both (for approximation by both). Per shuffle iteration, we calculated the explained variance, which yielded a shuffled distribution per prediction (e.g. *Figure 2f*). A prediction was defined as significantly above chance when the real explained variance was at least 2 SDs away from the shuffled distribution mean (corresponding to $p<0.05$).

## Heterogeneity calculation

We calculated heterogeneity of population activity as follows (see also *Figure 3d*). For each independent data source $i$ (i.e. a neuron) that provides a certain measurement $R$ at each time point t (i.e. $dF/F_0$ activity of a single trial), we first z-scored the responses of $i$ over all trials $T$ (i.e. all contrasts and orientations). For all analyses we took $t$ to be a single trial, except those shown in *Figure 5*, where $t$ corresponds to a data acquisition point (i.e. a single calcium imaging frame), and calculated heterogeneity as follows. First, we z-scored all trial responses per neuron over all trial types (therefore high-contrast, preferred orientation stimuli yield higher z-score values than low-contrast, non-preferred orientations):

$$Z_{i,t} = (R_{i,t} - \mu_i)/\sigma_i \qquad (2)$$

**Z** is therefore a matrix containing $n$ (number of neurons) by $T$ (trials) measurements of standard deviations ($\sigma$) from the mean over all trials ($\mu$). Next, for each trial $t$, we calculated the pairwise distance (in standard deviations) from each independent source to each other independent source (pairwise neuronal $\Delta\sigma$): we repeated the z-scored population response vector $\mathbf{z}_t$ over its singular dimension $n$ times, where $n$ is the number of neurons in $\mathbf{z}_t$ (yielding a square matrix), subtracted this

matrix from its own transpose $\mathbf{z}_t^T$, and took the absolute of the result, giving the heterogeneity matrix $\mathbf{H}_t$:

$$\mathbf{H}_t = |\, \mathbf{z}_t - \mathbf{z}_t^T \,| \tag{3}$$

To get a single measure of population heterogeneity per trial ($h_t$), we next took the mean of all z-scored distances between neuronal pairs ($i,j$) in the heterogeneity matrix; this provides a measure of the mean distance in activation levels within our population at a single trial $t$:

$$h_t = \sum_{i=[1 \ldots n-1]} \sum_{j=[i+1 \ldots n]} (H_{t,i,j}) / ((n \cdot (n-1))/2) \tag{4}$$

## Effect size of mean population dF/F$_0$ and heterogeneity

We used a measure of effect size using Cohen's $d$ to quantify which metric (mean dF/F$_0$ or heterogeneity) showed a stronger correlation with visual detection. We calculated for both metrics per animal the effect size for all intermediate contrasts (0.5–32%) between hit and miss trials and took the mean over these four values, yielding a mean hit/miss effect size for dF/F$_0$ and heterogeneity per animal. This allowed us to perform a paired t-test between the dF/F$_0$ effect sizes and heterogeneity effect sizes to test for statistical significance. Cohen's $d$ is defined as the difference between the two means (hit; $\mu_h$, miss; $\mu_m$) divided by the pooled standard deviation for the data:

$$d = (\mu_h - \mu_m) / \sigma_p, \tag{5}$$

where $\sigma_p$ is defined as

$$\sigma_p = \sqrt{[(n_h - 1) \cdot \mathrm{var}_h + (n_m - 1) \cdot \mathrm{var}_m] / (n_h + n_m - 2)} \tag{6}$$

## Instantaneous Pearson-like correlations and sliding-window Pearson's correlations

For a pair of neurons $x$ and $y$, the Pearson's correlation ($R$) of their activity can be calculated by z-scoring each neuron's response vector (as in *Equation 2*) and taking the mean of the element-wise multiplication of the two vectors:

$$R_{x,y} = \sum_{t=[1 \ldots T]} (Z_{x,t} \cdot Z_{y,t}) / T \tag{7}$$

Here, notations are the same as for *Equations 3–5*; $t$ is a single trial and $T$ is the total number of trials. Using this equation, it is impossible to obtain an instantaneous correlation value between two neurons for each trial because its calculation requires taking the mean over all trials. This poses a problem if we want to estimate the instantaneous correlation value between a pair of neurons for a given trial. Therefore, we computed a modified measure, the instantaneous Pearson-like correlation ($\check{R}$). For each pair of neurons, we calculated the z-scored element-wise product (each element being a single trial), which yields a three-dimensional matrix $\check{Z}$ with size [$n$ by $n$ by $T$], where $n$ is the number of neurons:

$$\check{Z}_{x,y,t} = Z_{x,t} \cdot Z_{y,t} \tag{8}$$

Taking the mean over the matrix's third dimension (trials) gives the conventional Pearson's pairwise correlation matrix over neuronal pairs. However, the matrix also allows us to approximate the mean pairwise correlations within the whole population at any given trial ($\check{R}_t$) by taking the mean over all unique neuronal pair values in matrix $\check{Z}$:

$$\check{R}_t = \sum_{i=[1 \ldots n-1]} \sum_{j=[i+1 \ldots n]} (\check{Z}_{i,j,t}) / ((n \cdot (n-1))/2) \tag{9}$$

Similarly, we can take the standard deviation instead of the mean over all unique pairs per trial to estimate the spread of the instantaneous pairwise correlation distribution. However, note that while the instantaneous Pearson-like correlation is similar to the conventional Pearson correlation, $\check{R}$ is not bounded within the interval [−1 1], because the z-scored element-wise product and the mean-operator work over different sets of values (i.e. matrix dimensions).

We additionally used for comparison a more conventional measure of correlations across time by using a wavelet-based sliding-window correlation (*Cooper and Cowan, 2008*). The time scale of the wavelet used in all sliding-window analyses was set to 1.0 s as this was similar to the animals' median reaction times and should therefore maximize the stimulus-driven change in neuronal pairwise correlations.

## ROC analysis of hit/miss separability

We quantified the single-trial behavioral response predictability using an ROC approach by calculating the area under the curve (AUC) for a false positive rate versus true positive rate plot. All ROC curves were computed separately per contrast and animal for both heterogeneity and mean population dF/$F_0$ (*Figure 3g*). For comparison across animals, we averaged the AUC of the four test contrasts per animal, yielding a single AUC value per animal for both heterogeneity and dF/$F_0$ (*Figure 3h*).

## Decoding of stimulus presence

To ascertain the performance of a decoder on the same task as we required the mouse to perform, we created an algorithm that calculated the probability of a stimulus being present. This decoder was based on a previously published maximum-likelihood-naive Bayes decoding algorithm (for a more complete description, see *Montijn et al., 2014*). For each neuron and stimulus orientation, we computed the mean and standard deviation of mean dF/$F_0$ during presentation of a 100% contrast stimulus as well as the mean and standard deviation during 0% probe trials. For each test trial and neuron with the preferred orientation as the trial's stimulus orientation, we calculated the probability a stimulus was present by reading out the likelihood density function for 0% and 100% contrast trials. The product over neurons in the preferred population for each trial then yields a population posterior probability value for stimulus absence (0% likelihood) and presence (100% likelihood). The decoder's read-out was the posterior with the highest probability. Because the likelihood was only based on 0% and 100% contrast responses, automatic cross-validation was ensured for decoding test contrast stimuli. After decoding stimulus presence for all trials, we split the trials into hits and misses and calculated the percentage for which the decoder indicated a stimulus was present per response type and contrast, averaging over repetitions and orientations. This yielded two curves per animal (see *Figure 4d*). We tested for statistically significant differences between response and no-response trials by performing a paired t-test over animals on the intermediate contrasts (0.5–32%).

Furthermore, we quantified the similarity of our decoder's performance to the animal's performance in the visual stimulus detection task by calculating the similarity per animal of its actual behavioral performance to the decoder's performance (Pearson's correlation over contrasts). We compared this value to the similarity obtained with a bootstrapped shuffling procedure (1000 iterations). Here, we shuffled the animal's behavioral and decoder performance over contrasts, recalculated the similarity index, and took the mean over all iterations as the resultant shuffled similarity. To test for statistical significance, we performed a paired t-test over animals between the shuffled and real similarities (*Figure 4e*).

Moreover, we investigated the similarity between the animal's and decoder's output at a single-trial level with a chi-square analysis. Pooling all trials across animals showed significant correspondence between the decoder and animal's judgment of stimulus presence; hit trials were more often decoded as 'stimulus present' and miss trials more often as 'stimulus absent' (*Figure 4—figure supplement 2j*). Note that this decoding procedure is not optimal; the absolute decoding performance therefore should not be interpreted as reflecting the actual amount of information present in the neural responses. The purpose of this decoder is merely to test—in coarse terms—the similarity between the neural signal and the animal's behavior.

## Behavioral response predictability

We analyzed the predictability of behavioral responses before they occurred based on either the mean population dF/$F_0$ response or population heterogeneity between 3 and 0 s before stimulus onset (*Figure 5e*). Hit trials were split into the 50% fastest and 50% slowest reaction times per contrast per animal and then averaged over contrasts, yielding 6 data points per animal: the mean pre-stimulus population dF/$F_0$ and mean population heterogeneity preceding fast, slow. and miss trials.

We then quantified the consistency of differences over animals by calculating the distance of these points per animal to the mean of their own response group and the other two. We defined the predictability metric per point $i$ (animal) for two response types $r1$ and $r2$ (i.e. two types out of fast, slow, or miss) as

$$\delta_{r1,r2,i} = ((\|d(i_{r1},\mu_{r2})\| / (\|d(i_{r1},\mu_{r2})\| + \|d(i_{r1},\mu_{r1}^{\neg i})\|)) - 0.5) \cdot 2,  \quad (10)$$

where $\|d\|$ is the absolute Euclidian distance (vector magnitude), $\mu_r$ is the mean location of $I_r$– where $I_r$ is the group of points for response $r$ – and $\mu_r^{\neg i}$ indicates the mean location of $I_r$ without point $i$. This analysis yields a vector $\delta_{r1,r2}$; the separability between response type $r1$ and $r2$. Random placement would lead to a separability of $\delta = 0$, so we quantified statistically significant predictability of responses by performing FDR-corrected one-sampled t-tests (vs. 0) for each separability vector and both neuronal population metrics (heterogeneity and mean dF/$F_0$). We also tested whether the separability was higher for heterogeneity or dF/$F_0$ by performing FDR-corrected paired t-tests between dF/$F_0$ and heterogeneity separability vectors for the same response type comparisons (*Figure 5e*).

## Rise time to maximum heterogeneity

We defined the rise time to maximum stimulus-driven heterogeneity as the time it took the population heterogeneity to rise from 10% to 90% of the difference between pre-stimulus baseline levels and maximum heterogeneity during the stimulus period. This rise time was calculated on the mean curves per animal and contrast as shown in *Figure 5d*. To create the graph shown in *Figure 5f*, we took the rise time across test contrasts per animal (n=8) and behavioral response type (miss, slow, fast). We tested for significant differences in average rise times between response types with paired t-tests across animals.

## Population activation pattern consistency

Detection of a visual stimulus might be associated with consistencies in population activity. We, therefore, analyzed whether the inter-trial correlation of population activity varies depending on the behavioral performance of the animal. We again separated fast, slow, and miss trials, and for each stimulus orientation calculated the correlation of the dF/$F_0$ response vector between pairs of trials with the same type of behavioral response (*Figure 6a*). We separated the neuronal responses for that orientation's preferred and nonpreferred population of neurons, also to address whether consistency across trials might be restricted to the preferred population or would also occur in the nonpreferred population (*Figure 6d–d*). Note that because we calculated the correlations separately for preferred and nonpreferred populations, the relative contribution of the orientation signal is fairly low, which explains the relatively low correlation values. To assess above-chance similarities, we compared these values to correlations obtained from shuffled data. By shuffling within each stimulus orientation all trial identities randomly for each neuron, the orientation signal is preserved, but other similarities across trials are destroyed. We repeated this shuffling procedure 100 iterations and took the mean of these 100 iterations as shuffled correlation value per animal (*Figure 6b–d*). To test for statistically significant consistencies in population activation patterns, we performed FDR-corrected paired t-tests between the real and shuffled correlation values over animals for the different response types and the two neuronal population types. We also quantified the differences between response groups in the real data with paired t-tests (miss vs. slow, miss vs. fast, and fast vs. slow).

## Analysis of multidimensional inter-trial distance in neural activity

To study the theoretical implications of our results relating to heterogeneity, we proceeded with an analysis of the question whether heterogeneity forms a special case of population codes that do not merely reflect an increased activity of all neurons upon visual detection. For the specific purpose of these analyses (shown in *Figure 7*), we use as definition for multidimensional heterogeneity the distance in neural space from the population's activity to the closest point on the main diagonal (see text and below for further explanation). Although this definition is computationally different from our pairwise definition of heterogeneity, it also captures the overall dissimilarity of responses within a population of neurons. Moreover, applying this procedure to z-scored dF/$F_0$ values yields Pearson's correlations of r > 0.9 when compared with our original definition of heterogeneity (*Equations 3 and 4*) and gives very similar hit/miss Cohen's d values (*Figure 3—figure supplement 2*).

The two metrics, therefore, likely capture the same neural phenomenon and show that heterogeneity can be studied by different, but related computational definitions.

To assess the distribution of neuronal population activity in multidimensional neural response space (where each dimension represents the activity of a single neuron; see *Figure 7a–d*), we calculated the inter-point distance (each point representing the population activity during a single trial) between all hit trial pairs and between all miss trial pairs. The distance in neuronal activity for a population of $n$ neurons between a pair of trials $x$ and $y$ in multidimensional space can be calculated as the $n$-dimensional Euclidian:

$$d\left(\mathbf{x},\mathbf{y}\right)=\sqrt{\left(x_1-y_1\right)^2+\left(x_2-y_2\right)^2+\ldots+\left(x_n-y_n\right)^2} \tag{11}$$

The pairwise inter-point distance is then given in units of neural activity (dF/F$_0$, *Figure 7e*). Note that this formula can also be used to calculate the multidimensional heterogeneity, as defined above, by taking the distance between any trial ($x$) and the closest point on the diagonal ($y$).

Next, we investigated the symmetry of population responses around the main diagonal as this symmetry gives an indication of whether heterogeneity is an epiphenomenal observation or a fundamental neural characteristic underlying visual detection (see text). In order to do so, we mirrored each point across the diagonal and recalculated the inter-point distances for the mirrored data. Mirroring across the diagonal was achieved by direct inversion of the signs per neuron relative to the main diagonal. For a population response $r = [r_1\ r_2\ \ldots\ r_i\ \ldots\ r_n]$, where $n$ is the number of neurons, the mirrored version $r' = [r'_1\ r'_2\ \ldots\ r'_i\ \ldots\ r'_n]$ was calculated as follows:

$$\mathbf{r}'=\left(\mu_r-\left(\mathbf{r}-\mu_r\right)\right) \tag{12}$$

where $\mu_r$ is the mean population response over $r$.

## Removal of mean and heterogeneity, and subsequent hit/miss decoding

For the analyses displayed in *Figure 7g,h*, we removed the mean and/or heterogeneity from the population responses and assessed the effect on decoding accuracy of hit/miss responses during test contrast stimuli. As mentioned before, for these analyses heterogeneity was defined as the distance to the main diagonal. As such, removal of the mean without influencing heterogeneity is trivial and can be achieved by simply subtracting the mean population response from all neuronal dF/F$_0$ values obtained for each trial. Briefly, heterogeneity was removed from each trial without affecting the mean in two steps; first heterogeneity was removed, and next any influence on the mean was remedied by adding the difference between the new and old mean. First, heterogeneity was removed by dividing each neuron's response during that trial by the square root of the sum of the squared differences between the neuronal responses and the mean (i.e. by dividing by the heterogeneity):

$$\mathbf{r}'^{\neg \mathbf{H}}=\frac{\mathbf{r}}{\sqrt{\left(r_1-\mu_r\right)^2+\left(r_2-\mu_r\right)^2+\ldots+\left(r_n-\mu_r\right)^2}} \tag{13}$$

Next, changes in the mean were corrected by removing the new mean of the heterogeneity-removed population activation ($\mu_{r'^{\neg H}}$) and adding the old population mean $\mu_r$:

$$\mathbf{r}^{\neg \mathbf{H}}=\mathbf{r}'^{\neg \mathbf{H}}+\mu_r-\mu_{r'^{\neg H}} \tag{14}$$

This way, the heterogeneity (i.e. the Euclidian distance of that trial's population activity to the main diagonal) is normalized to 1.0 for all trials. The multidimensional location relative to the diagonal is preserved, but its distance is always the same; all trials now fall on a cylinder with a radius of 1.0 dF/F$_0$ around the main diagonal. In other words, the population activation during a trial is projected as a vector from the closest point on the diagonal to the trial's position, and the vector's angle is preserved, but its magnitude is normalized to 1.0. Both properties (mean and heterogeneity) can be removed by subtracting the mean from the heterogeneity-removed responses. Removing the mean as well as the heterogeneity collapses this cylinder onto a circle through multidimensional space around the origin.

## Control analyses for confounds related to running, licking, and eye movements

To control for potential locomotor confounds, we split all data sets into trials where the mouse was still (90.9% ± 3.6% of trials) and where it was moving during stimulus presentation (8.1% ± 3.6% of trials), and reanalyzed our data. Our results with exclusion of running trials (*Figure 2—figure supplement 1g,h*) are very similar to our original analysis (*Figure 2a,b*), showing that the effects we observed cannot have been due to running-induced modulations (paired t-test, hit vs. miss, 0.5–32%, p<0.05).

Another potential confound for our results could be that response trials induce signals related to motor feedback or motivation to initiate motor actions because the animal initiates licking as a behavioral response. This also seems unlikely; because 0% contrast probe trials did not induce neuronal activity during false alarms (*Figure 2—figure supplement 1a*, green line). Theoretically, however, such signals could still be present and influence population activity only when occurring concurrently with visual stimulation. To control for this, we re-performed our analyses shown in *Figure 2a,b*, but now used data only from the first 0.4 s after stimulus onset; approximately 0.8 s before the mean reaction time. Leaving a window of 0.8 s between the latest frame included in the data analysis and the licking response should also eliminate potential modulatory activity from motor cortex related to the preparation of licking. The results from this control analysis were slightly noisier due to the shorter data acquisition duration per trial, but showed no qualitative differences to the original analysis regarding heterogeneity (*Figure 2—figure supplement 1i,j*). The intermediate contrasts still showed significant enhancements in heterogeneity (p<0.01) during hit trials, but we found no significant differences for mean population $dF/F_0$ (p=0.543). We, therefore, conclude that our results regarding heterogeneity are not confounded by motor-related modulations due to running or licking, nor by reward-expectation prior to licking responses, and confirm that the mean population $dF/F_0$ is not or less useful as a measure of neural correlates of perception.

To control for possible effects of blinking and saccades, we performed pupil detection on our eye-tracking data and removed all trials in which the animals blinked or made saccades during any time of the stimulus presentation [10.2% ± 4.6% of trials removed (mean ± SD)]. We re-performed our analyses on only the trials where no contamination by incorrect eye position and/or closing of the eyelids was possible (*Figure 2—figure supplement 1k,l*) and observed that our results regarding heterogeneity were qualitatively and quantitatively similar to our original analyses, but that the $dF/F_0$ results were again more sensitive to a conservative analysis (hit/miss difference for intermediate contrasts, paired t-test, n=8; $dF/F_0$, p=0.136; heterogeneity, p<0.005). We conclude that our main results are not biased by incorrect eye position and blinking.

## Orientation decoding

We addressed whether the orientation information contained in the population responses was dependent on the mean $dF/F_0$ and heterogeneity during stimulus presentation. We decoded the presented stimulus orientation for each contrast separately (i.e. 100% contrast trials based on likelihood from 100% contrast trials, etc.) by a leave-one-out cross-validation and afterwards split all trials into correctly and incorrectly decoded ones (*Figure 4—figure supplement 2b*). To quantify the dependence of decoding accuracy on $dF/F_0$ during stimulus presentation, we took for each contrast the trials with highest and lowest 50% of $dF/F_0$ and calculated the mean decoding accuracy for both groups (high and low activity). Next, we took the mean for these groups over contrasts per animal and calculated a percentage decoding accuracy increase for the highest versus lowest 50% $dF/F_0$ trials (see *Figure 4—figure supplement 2c*). To test for statistical significance, we performed a one-sample t-test of the percentage increase values over animals. For heterogeneity, we performed the same steps and performed a t-test versus 0% increase (*Figure 4—figure supplement 2c*).

## Stimulus feature decoding

To address whether visual stimulus features (i.e. orientation and contrast) were more accurately represented by neuronal population activity during correct versus incorrect behavioral performance, we used a Bayesian maximum-likelihood decoder as previously described to extract those features from the population activity (for a more complete description, see *Montijn et al., 2014*). We defined all combinations of orientations and contrasts as different stimulus types, yielding a total of 21 different

stimulus types (four orientations times five contrasts plus probe trials). Next, we performed a leave-one-out cross-validated decoding procedure for all trials and calculated the mean percentage correct decoding trials for hits and misses per stimulus type; then we averaged the percentage correct over stimulus types, yielding an accuracy per animal for hit and miss trials. We tested for a statistical difference between hits and misses with a paired t-test over animals (*Figure 4—figure supplement 2a*).

## Noise correlations

To investigate detection-related increases or decreases in noise correlations (*Figure 4—figure supplement 2f,g*), we first calculated a response vector for each stimulus orientation θ that was presented during a test contrast trial. Here, each element in the vector is the neuron's response to a single presentation $t$ (i.e. a trial) of that stimulus orientation:

$$\boldsymbol{R}_\theta = [R_{\theta_t} \dots R_{\theta_n}] \tag{15}$$

where $n$ is the number of repetitions per response type per orientation. Because we aim to compare a single noise correlation value per neuronal pair $i,j$, we took the mean noise correlation over all four stimulus orientations:

$$\rho_{i,j}^{noise} = \frac{\sum_{\theta=0}^{135} \mathrm{corr}(\boldsymbol{R}_{i,\theta}, \boldsymbol{R}_{j,\theta})}{4} \tag{16}$$

The noise correlation is, therefore, an index of the mean trial-by-trial variability shared by pairs of neurons over all stimulus orientations.

## Behavioral response predictability on single-trial basis

To verify that the behavioral predictability before stimulus onset that we found (*Figure 5e*) was not merely a group-level effect, but was indeed also a single-trial phenomenon, we subsequently performed single-trial decoder-based predictions of fast/slow/miss behavioral responses that occurred during the subsequent stimulus presentation (see *Figure 5—figure supplement 1*). We used a similar leave-one-out cross-validated naive Bayes decoder as described above for fast, slow, and miss trials, and calculated per trial the relative likelihood that the subsequent stimulus presentation would lead to a miss, fast, or slow response. We then split the predictive decoding results per actual behavioral response group and averaged the relative prediction likelihood per animal. This yields three relative probability values per actual response type per animal. Assigning an angle to each of these behavioral responses that are separated by $2/3\pi$ on the unit circle and taking the relative likelihood as the vector magnitude, it is then possible to calculate a resultant prediction vector per actual response type per animal. To quantify statistical significance, we multiplied an angle-based correctness index (+1 when the resultant prediction vector angle is perfectly aligned to the actual response angle and –1 when they are separated by $1\pi$) with the vector magnitude, giving a normalized decoding accuracy index between –1.0 and +1.0, where chance level is 0. Lastly, we performed one-sample t-tests on the normalized decoding accuracy indices over animals and response types for heterogeneity and dF/F$_0$, and a paired t-test between dF/F$_0$ and heterogeneity (*Figure 5—figure supplement 1*).

## Acknowledgements

The authors thank Q Perrenoud, G Meijer, and M Vinck for feedback on earlier versions of the manuscript and fruitful discussions. They would also like to thank W Oldenhof, J Verharen, and L Forsman for assistance with training the animals.

## Additional information

### Funding

| Funder | Grant reference number | Author |
| --- | --- | --- |
| Nederlandse Organisatie voor Wetenschappelijk Onderzoek | Excellence grant for the Brain & Cognition project 433-09-208 | Cyriel MA Pennartz |

| | | |
|---|---|---|
| European Commission | EU FP7-ICT grant 270108 | Cyriel MA Pennartz |

The funders had no role in study design, data collection and interpretation, or the decision to submit the work for publication.

## Author contributions

JSM, Built the setup, performed the experiments and analyzed the data, designed the experiments and analyses, and wrote the paper; PMG, Built the setup; CMAP, Designed the experiments and analyses, and wrote the paper

## Author ORCIDs

Jorrit S Montijn, http://orcid.org/0000-0002-5621-090X

## Ethics

Animal experimentation: All experimental procedures were conducted with approval of the animal ethics committee of the University of Amsterdam (DED234). All animals were housed socially in enriched cages and received analgesia (buprenorphine) and anesthesia (isoflurane) during invasive operations to minimize suffering.

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
