## [Decision Letter]

Thank you for submitting your work entitled "Mouse V1 population correlates of visual detection rely on heterogeneity within neuronal response patterns" for peer review at *eLife*. Your submission has been evaluated by Eve Marder (Senior editor), a Reviewing editor, and three reviewers.

The reviewers have discussed the reviews with one another and the Reviewing editor has drafted this decision to help you prepare a revised submission.

The reviewers articulated a desire to see your responses to their critiques and felt you should be given a chance to revise your work. We are allowing you to submit a revision, as long as you understand that a positive outcome is not assured, and will depend on whether the reviewers and Reviewing editor feel that you have adequately revised your manuscript and/or rebutted the critiques.

Summary:

The statistics of population neuronal responses of early sensory cortices associated with animal perceptual behavior is an important issue in neuroscience. In this paper, Montijn and collaborators compared neuronal fluorescence signals from L2/3 mouse V1 with behavioral responses in a detection visual task. The reviewers think that there are interesting results but it is not clear up to what point the observed correlation between behavior and the heterogeneity measure supports the authors' claim. An important issue is that authors neglect many previous developments on the neuronal correlates in other early primary cortical areas (somatosensory cortex and auditory cortex). The paper focuses on the neuronal correlates of V1 with visual detection performance, but this is a general problem and the authors should mention this.

Essential revisions:

*Reviewer #1:*

1) The authors state that "for each trial took the responses of only neurons that preferred the presented stimulus orientation" (subsection “Response dissimilarity within neuronal populations correlates with detection”). This practice is extremely dubious. First, it is totally unclear how it affects subsequent analysis. Were Z-scores calculated once over all orientations or calculated for separately each orientation for those neurons that preferred it? When examining the relationship between behavior and heterogeneity, was a correlation calculated for each orientation or over all orientations?

2) The presented stimuli consist of square wave gratings with 8 different directions but a response to any orientation was rewarded. In essence, this becomes a matter of responding to a change in light level. Thus analyzing responses to oriented stimuli may result in a bias towards those neurons with inherent responses to the stimuli which obscures or masks the responses from the neurons actually involved in the discrimination of the change in intensity. Nonetheless, the authors, on the basis of Ca^2^ -transient measurements from ~100 L2 neurons in monocular V1, conclude that "visual perception does not correlate well with mean response strength, but is significantly correlated with population heterogeneity." This statement ought to be drastically revised to reflect that it is contingent on the ad-hoc procedures chosen by the authors, and how the correlation is calculated using data-selection procedures based on orientation, which was not part of the behavioral task.

3) Since the animal needs in principle only to respond to an increase in ambient light intensity brought about by the stimulus and since no behavioral dependence on orientation has been reported, all of the analysis concerning orientation selectivity (preferred populations etc.) is potentially irrelevant, and the logic behind this experimental design is not clear. If one were designing an experiment to test for a correlation between mean response strength and visual perception, surely it would be wise to do one's best to ensure that the neurons from which responses were recorded had response properties that were at least to some degree related to the discrimination target? While it would be equally unwise to assume that orientation selective neurons in V1 do not play a role in visual discriminations not involving oriented stimuli at their preferred orientation, the failure on the authors' side to discuss in any way the caveats associated with their experimental design and simultaneously to draw the conclusions that they do and state them as strongly as they do is remarkable.

4) The heterogeneity measure, the sum of pairwise absolute z-score differences, does not correspond to any normal usage of the word heterogeneity and is never adequately justified. For example, if all neurons respond to a given stimulus with the same fluorescence increase, the heterogeneity of that stimulus will not be zero but will depend on their responses to other stimuli. Even a trial that elicits no fluorescence change in any neuron the heterogeneity will not be zero. Since the measure is based on z-scores, it will amplify fluorescence noise in neurons that are less frequently active so that for sparse activity noise can dominate the measure, but this issue is never discussed. While it does indeed seem to correlate better than some other measures with behavior, the manuscript does not adequately explain how this measure was calculated and in any case this measure would not tell us what is going on the brain.

5) The alternative measure "instantaneous Pearson correlations" suffers from the same problems as "heterogeneity." It is improperly named as it is not a Pearson correlation. Time varying correlation measures already exist and should be mentioned; they are generally based on sliding windows (e.g. "Time-varying correlation coefficients estimation and its application to dynamic connectivity analysis of fMRI" Fu et al. 2013 or "The sliding window correlation procedure for detecting hidden correlations: existence of behavioral subgroups illustrated with aged rats" Schulz and Huston 2002).

6) The nature of the decoder used (subsection “Heterogeneity predicts reaction time”) is never explained in the main text or Methods. The extremely convoluted use of a similarity metric and p-value based on comparison to randomly shuffled data (Figure 4) to claim that the decoder and the animal behave similarly is not a clear and honest presentation of results. The similarity metric was not explained in the Methods. There is nothing to support the statement that "the performance as a function of contrast was strikingly similar to the animals' actual behavioral performance."

7) The assertion, in the Introduction, that "a widely held assumption in computational models of vision is that neurons in distributed cortical architectures have relatively fixed roles in information coding" is a straw-man argument. The authors do not adequately characterize what this assumption of "fixed roles" means, and also fail to characterize the diverse set of existing theories and conjectures about how the visual system may function.

8) We need to see much more raw data so as to evaluate data quality. In particular, we should see supplementary movies showing simultaneous raw, unprocessed imaging data, behavior, and "heterogeneity" for ~10 consecutive trials.

9) The very large responses of some neurons with nearly 100% DF/F in Figure 1 don't seem to match the very modest DF/F of 4% over "preferred populations" in Figure 2. Are the data in Figure 1 not representative of the full dataset? Or is the time window for averaging each trial's responses perhaps too long? The presentation, figure and analyses are unclear.

10) The first stated aim is to ask: “does visual detection correlate with mean visual response strength or other metrics?". This may be of interest if one could determine for certain that the response strength was being determined for the neurons really involved in the detection/perception required by the task. But why should we care what L2/3 is doing during this task, when it may not even be involved in generating the behavioral response?

11) The authors assert in the Introduction that "specific ensemble activation patterns reoccur across temporally spaced trials in association with hit responses, but not when the animal fails to report a stimulus." I do not understand how, on the basis of the data presented in Figure 6 and the manuscript text associated with it, that this conclusion can be drawn. The authors state: "We again split the data into miss, fast and slow response trials, and computed the correlations between response patterns from different trials separately for preferred and non-preferred neuronal populations…" What response patterns are being correlated? The Methods states that the "mean inter-trial correlations over animals" was compared. I find the link between this measure and the conventional definition of ensemble tenuous at best. Further, the calculated correlation coefficients are very low (<0.12), which does not support well the claim made above.

12) The authors describe their method for assessing the extent of slow drift in the z-plane, which they quantize into 10μm bins. It is unclear what additional effects this may have on the measured Ca^2^ -transients, something that would be best determined empirically using simultaneous electrophysiology. More importantly, fast shifts in the z-plane are a considerably larger problem, and these would be anticipated as the animal changes its posture or shifts fore- or hindlimb. This sort of "fidgeting" is commonly observed in advance of a rodent making a behavioural response. How the authors measured these postural adjustments is not clear, neither is the effect that these movements have on the activity recorded. It is certainly conceivable that a z-shift could move the focal plane further inside some neurons and further outside others, thereby increasing "heterogeneity."

13) Previous multiphoton Ca^2^ -imaging studies have shown that correcting xy-shifts uniformly across the whole image is not sufficient for motion correction in awake animals (see Dombeck et al. 2007, Greenberg and Kerr 2009). As described above, motion-associated artefacts resulting from the fidgeting of the animal around a response are not quantified and potentially important.

14) The caveat that the only neurons from which recordings were made were superficial neurons ought to have been explicitly discussed. Is it not conceivable that the correlation of mean activity with perception might be significantly higher for neurons in deeper layers?

15) How did the authors control for possible ocular torsion (twisting of the eye and retina round the optic axis) during the experiment? This would totally invalidate all analyses based on orientation if present but not accounted for.

*Reviewer #2:*

1) The concern is about the animal's behavior. The performances shown in Figure 1 are relatively low at 100% contrast; in many cases slightly different than the ones at 32%. The presence of errors at full contrast imply mechanisms other than visual detection contributing to the animal's response variability that will potentially contaminate all other conditions as well.

2) Regarding the correlation between heterogeneity and behavior, the authors claim that "…the increased spread of neuronal response strengths within a population determine the behavioral accuracy". This reviewer is concerned about how strong is the change in heterogeneity between hit and misses to support this claim. In his opinion the authors should explicitly quantify how predictive is the animal's decision from this population measure, on a trial-by-trial basis.

3) He finds very interesting the fact that the measure of heterogeneity – but not the mean population response – correlates with detection. However, as far as he understands, this would be the case in any situation in which the detection of the stimulus is represented by a population code that is not merely an increase of activity of all neurons. The mean population response is only one particular projection of the population activity (let's say, described by the vector [1 1 … 1]). If detecting the stimulus activates the neural population in any other direction in neural space, this measure of heterogeneity will increase (because some neurons increase activity while others decrease). In particular, if detecting or not the stimulus modulates the population activity in a direction orthogonal to [1 1 … 1], the mean population response will not be affected (and won't correlate with the animal's behavior). His concern is that, if this is the case, it is not heterogeneity per se that is relevant, but the presence of complex population patterns of activity that are not visible at the level of the mean response. He thinks the authors should check if there is a population signal other than the mean response that correlates with the animal's decision.

4) The authors claim that ensemble patterns reoccur upon presentation of the same stimulus. However, inter-trial correlations of population responses are relatively low (~0.11). They should explain what value they take as a reference to validate this claim and why. Correlations could increase because of reasons other than reoccurring of the same activity pattern; a more detailed analysis is needed to support this claim.

5) He believes it is necessary to explain why the authors chose this particular measure of spread in neural responses, as opposed to – arguably – more natural ones like the variance. If the variance does not correlate with behavior as much as heterogeneity does, then this might also be informative of the properties of the population code. A set of related statistics are examined in regard to reaction times (Figure 4) but not in relation to the decision of the animal.

*Reviewer #3:*

Positive points:

1) Evaluated multiple metrics for stimuli detection.

2) Propose a new metric for population heterogeneity, where dissimilarly activated neurons have high population heterogeneity.

3) Data from a sufficient number of mice, 8, were collected and analyzed and the results hold across animals.

Negative points:

1) Preferred orientation and non-preferred orientation neurons are analyzed separately – this ignores potential interactions between neurons (subsection “Data processing”).

2) The preferred orientation neurons are selected using the mean dF/F0 value, however, the main result of the paper suggests that a different metric, heterogeneity, is more robust in capturing stimuli recognition; how will the analysis be affected if the same metric is used for pruning the neurons? (subsection “Calculation of preferred stimulus orientation”).

3) As defined, heterogeneity seems a reasonable metric, however, it only considers pairwise relationships between neurons; a more holistic, group-level metric should be considered, since the goal of the analysis is to discover groups of neurons.

4) Can you explain or cite the reasoning behind using the procedure in the subsection “Behavioral response predictability on single trial basis”, to compute a prediction? Can the model likelihood be used to make predictions instead?

[Editors' note: further revisions were requested prior to acceptance, as described below.]

Thank you for resubmitting your work entitled "Mouse V1 population correlates of visual detection rely on heterogeneity within neuronal response patterns" for further consideration at *eLife*. Your revised article has been favorably evaluated by David Van Essen (Senior editor) and two reviewers. The manuscript has been improved but there are some remaining issues that need to be addressed before acceptance, as outlined below:

In brief, both reviewers had positive comments about the revisions but also request minor additional revisions that will not require re-review.

Reviewer #1:

Regarding the authors’ response to Reviewer #1, comment 10: The study by Glickfeld activated PV neurons in a 1mm diameter around the injection pipettes ~up to 1mm below the V1 surface and showed that this increased the threshold for detection of both orientation and contrast by the animals. I do not see the relationship between the author's response to my question and the question. Why is it reasonably assumed that L2/3 is involved in the task that is presented?

Regarding the authors’ response to Reviewer #1, comment 11: Please change the last sentence in the Abstract to reflect the changes in terminology by removing ensembles (see below). I’m not sure what “selective and dynamic neuronal ensembles” are. Please also rephrase the first paragraph of the Discussion, which suffers from the same issue.

From the Abstract:

"Contrary to models relying on temporally stable networks or bulk-signaling, these results suggest that detection depends on transient activation of selective and dynamic neuronal ensembles."

*Reviewer #2:*

Single-trial population recordings in behaving animals have the potential to uncover how the dynamics of a network of neurons give rise to perception, decision and behavior. In the context of visual detection, given the activity of a population of neurons, what is the population measure that better relates with the animal detecting or not the stimulus is unknown. This study shows that in L2/3 of primary visual cortex, measures of spread of neural activity are more predictive of the animal's detection than mean-based measures. The authors did a very good job addressing the issues mentioned in the revision. I believe the paper has improved significantly both in the analysis of the data and in the precision with which the claims are expressed.

Response to my prior comments:

1) I had noted that the low performances at full contrast imply mechanisms other than visual detection contributing to the animal's decision (lack of motivation, for example). This means that test contrast trials are probably contaminated with a significant amount of trials (close to 50% for several animals) in which the animal actually detected the stimulus but didn't respond. The authors argue that heterogeneity does not reflect these other mechanisms because it's equal for both behavioral responses at full contrast. I agree with the argument and understand that the low performances might actually be diluting the effect reported in the paper. But I still would like to ask, does the distribution of heterogeneity in "No Resp" trials show any hint of bimodality, reflecting the 50% of trials in which the stimulus was in fact detected?

2) I had requested a quantification of how predictive is the single-trial value of heterogeneity of the animal's behavior. This was added in Figure 3.

3) I had asked whether the reported effect of increased heterogeneity could be an artifact of the presence of complex -but well-defined- patterns of activation orthogonal to the mean activity. The authors developed an elegant new analysis to address this question by mirroring neural responses with respect to the mean and measuring its symmetry. The results show that neural responses are a bit asymmetrical, pointing to the existence of a structured activation related to visual detection, although the effect size is very small. Besides, this analysis leads to the finding that hits are more structured than misses. Finally, removal of the mean, the heterogeneity or both, allows identifying the importance of each property on hit/miss decoding. I consider the point well taken.

4) I had requested more details on the analysis of reoccurring patterns of activity between trials. The authors addressed this question by expanding the analysis of correlations between population patterns and added the corresponding controls.

5) I had asked for a deeper explanation of why they choose this particular mathematical definition for heterogeneity as opposed to others. The authors expanded the analysis of hit/miss difference for other metrics of heterogeneity and found that many lead to the same results. They mention this fact in the revised manuscript, clarifying that the main result is that measures of "spread" of neural responses are more predictive than mean-based ones.

---

## [Author Response]

*Essential revisions:*

Reviewer #1:

*1) The authors state that "for each trial took the responses of only neurons that preferred the presented stimulus orientation" (subsection “Response dissimilarity within neuronal populations correlates with detection”). This practice is extremely dubious. First, it is totally unclear how it affects subsequent analysis. Were Z-scores calculated once over all orientations or calculated for separately each orientation for those neurons that preferred it? When examining the relationship between behavior and heterogeneity, was a correlation calculated for each orientation or over all orientations?*

We have clarified in the revised version of the manuscript that the line the reviewer quotes referred only to Figure 2; the reviewer’s concern about “dubious” practice was therefore based on an apparently unclear description in the original version and not on an actually dubious practice. We have more clearly stated in the revised version that we include all neurons in almost all analyses (subsection “Response dissimilarity within neuronal populations correlates with detection”, first, second and fourth paragraphs). We apologize for the confusion and hope these changes are sufficient to avoid further ambiguity. We also agree with the reviewer that the z-scoring procedure that heterogeneity is based upon could have been described more clearly, especially in the initial in-text description. We have edited this description (in the third paragraph of the aforementioned subsection) as well as in the Material and methods describing the calculation of heterogeneity (subsection “Heterogeneity calculation”).

To explain these clarifications in the manuscript, and to answer the reviewer’s questions more directly: the z-scoring procedure is performed on trial responses, not on single time points (except for Figure 5). Because heterogeneity calculations are trial-based rather than single-frame-based, we changed “time-point” to “trial” (subsections “Response dissimilarity within neuronal populations correlates with detection” and “Predictability of hit-modulation by consistent neuronal responses and whole-population fluctuations”). The z-scoring is performed for each neuron across all trials of all contrasts and orientations. Thus, trials of the neuron’s preferred, as well as its non-preferred, orientation were included in the z-score calculation. As such, each trial yields a heterogeneity value that includes the response of preferred and non-preferred neurons. We pooled the heterogeneity values across all trials, and calculated the mean heterogeneity for response vs. no-response trials (Figure 3), the mean for fast, slow and miss trials (Figure 5), and the correlation of heterogeneity with the animal’s reaction time (Figure 4). Moreover, to allow a more faithful comparison between the heterogeneity and mean dF/F0, we calculated both metrics using the whole population, the preferred population, and only the non-preferred population and show these results side-by-side in the original version as well as the revised version (Figure 3), with single animal examples as well as the mean over animals (Figure 3—figure supplement 1). These figures show that heterogeneity hit-miss differences are larger than mean dF/F0 hit-miss differences in each of these cases.

*2) The presented stimuli consist of square wave gratings with 8 different directions but a response to any orientation was rewarded. In essence, this becomes a matter of responding to a change in light level. Thus analyzing responses to oriented stimuli may result in a bias towards those neurons with inherent responses to the stimuli which obscures or masks the responses from the neurons actually involved in the discrimination of the change in intensity. Nonetheless, the authors, on the basis of Ca^2^ -transient measurements from ~100 L2 neurons in monocular V1, conclude that "visual perception does not correlate well with mean response strength, but is significantly correlated with population heterogeneity." This statement ought to be drastically revised to reflect that it is contingent on the ad-hoc procedures chosen by the authors, and how the correlation is calculated using data-selection procedures based on orientation, which was not part of the behavioral task.*

As the reviewer mentions, the task our animals had to perform was stimulus detection, and not orientation discrimination. However, because our goal was to investigate neural correlates of visual stimulus detection rather than orientation discrimination (as we state throughout the manuscript), we do not believe this presents a confound for our conclusions. In the original (as well as the revised) version we consistently used the term “detection” rather than “discrimination” throughout the manuscript. In the Introduction and Discussion we tried to place our results in a broader perspective and therefore referred to “perception” as a more general phenomenon. However, we recognize that our results are not based on a generalized, multi-faceted perception task, but on a visual stimulus detection task, and have therefore edited the manuscript to state instead that “visual stimulus detection does not correlate well with mean response strength, but is significantly correlated with population heterogeneity” (Introduction). Moreover, as we also mention below in response to a different comment by the reviewer, in the revised version we now explicitly discuss that our observations relate only to L2/3, and that L5 might show a different relationship with the detection of visual stimuli (subsection “Possible cortical layer specificity of poor correlation of mean population responses”).

*3) Since the animal needs in principle only to respond to an increase in ambient light intensity brought about by the stimulus and since no behavioral dependence on orientation has been reported, all of the analysis concerning orientation selectivity (preferred populations etc.) is potentially irrelevant, and the logic behind this experimental design is not clear. If one were designing an experiment to test for a correlation between mean response strength and visual perception, surely it would be wise to do one's best to ensure that the neurons from which responses were recorded had response properties that were at least to some degree related to the discrimination target? While it would be equally unwise to assume that orientation selective neurons in V1 do not play a role in visual discriminations not involving oriented stimuli at their preferred orientation, the failure on the authors' side to discuss in any way the caveats associated with their experimental design and simultaneously to draw the conclusions that they do and state them as strongly as they do is remarkable.*

We agree with the reviewer that the features of the drifting gratings were irrelevant for the performance of the task by our animals; this was also clear from the task description in the original version. However, we hypothesized that despite this irrelevance of stimulus feature specifics, correct stimulus detection would rely on general population phenomena that also influence coding fidelity of irrelevant stimulus features. To address the issue the reviewer raises here we performed the analyses described in the original version; we investigated whether there was a correlation between stimulus feature (contrast and orientation) representation fidelity by the recorded neurons and behavioral response (Figure 4—figure supplement 2). Although stimulus feature representation was irrelevant for the task the mice had to perform, we did find a behavioral correlate. We interpret these results as suggesting that the neural mechanisms governing performance in our behavioral task are also reflected in changes in population representation of irrelevant stimulus features. Although we consistently used “stimulus detection” rather than “orientation discrimination” throughout the manuscript, this line of reasoning was indeed insufficiently explained. We changed the text in the relevant section to now state that the stimulus features used for decoding (i.e. orientation) were indeed in fact irrelevant for the mouse to perform the task (subsection “Heterogeneity predicts reaction time”, third paragraph). However, note that during the inter-trial interval the screen’s background was isoluminant grey with the drifting gratings. Although not gamma-corrected, the grey background luminance was therefore in-between the perceived brightness of the black and white bars of the drifting grating stimuli.

Performing similar analyses as we did, but on a different behavioral task where mice have to discriminate orientations, would probably also yield interesting results. However, such an experiment addresses a different question than we focus on in our current manuscript. We believe that the observation that visual detection is correlated with increased fidelity of the neural representation of stimulus features, despite these features being irrelevant to the task, is interesting by itself. In particular, it suggests that sensory detection is coupled to better features representation than non-detection, even if that feature is task- irrelevant. This observation could not have been made if we had performed a similar experiment with an orientation discrimination task. We have added a similar comment as described here to the manuscript’s Discussion section (subsection “Comparison with other studies and neural interpretation of heterogeneity”, first paragraph).

*4) The heterogeneity measure, the sum of pairwise absolute z-score differences, does not correspond to any normal usage of the word heterogeneity and is never adequately justified. For example, if all neurons respond to a given stimulus with the same fluorescence increase, the heterogeneity of that stimulus will not be zero but will depend on their responses to other stimuli. Even a trial that elicits no fluorescence change in any neuron the heterogeneity will not be zero. Since the measure is based on z-scores, it will amplify fluorescence noise in neurons that are less frequently active so that for sparse activity noise can dominate the measure, but this issue is never discussed. While it does indeed seem to correlate better than some other measures with behavior, the manuscript does not adequately explain how this measure was calculated and in any case this measure would not tell us what is going on the brain.*

The definition of heterogeneity as the sum of pairwise differences in neuronal response might indeed come across as a narrow and specific form of heterogeneity, although its calculation was described in detail in the original version of the manuscript. As we have noted above in response to the reviewer’s first comment, some details may have been described somewhat unclearly, which we have rectified in the revised version (subsections “Response dissimilarity within neuronal populations correlates with detection” and “Predictability of hit-modulation by consistent neuronal responses and whole-population fluctuations”).

To address (among other things) the reviewer’s concerns about the specificity of our computational analysis, its relation to a more common definition of “heterogeneity” and the implication of our results for the functional properties of cortical circuits, we have added extra analyses using an alternative, more population-centered definition of heterogeneity (Figure 7, Results subsection “Analysis of heterogeneity in multidimensional space”). This alternative definition of heterogeneity is based on the location of the population activity in multidimensional neural response space, where heterogeneity is orthogonal to the main axis along which the mean population response changes (see the aforementioned subsection and subsections “Analysis of multidimensional inter-trial distance in neural activity” and “Removal of mean and heterogeneity, and subsequent hit/miss decoding”). We found a high trial-by-trial correspondence between both forms of heterogeneity, supporting the idea that the effects we observed in our data are robust and do not depend strongly on the precise computational definition of heterogeneity.

Regarding the reviewer’s comment on z-scoring of neural activity across trials, and the resulting non-zero heterogeneity arising from zero change in actual fluorescence, we would like to note that this is an effect of z-scoring neuronal activity and not of our pairwise-based definition of heterogeneity. The reviewer’s issue therefore comes down to whether z-scoring of neuronal responses is an acceptable practice. As the ability to separate two classes (which can also be the absence and presence of a stimulus) depends on the relative distance in standard deviations between the two distributions representing the responses to their respective stimulus conditions, z-scoring is a natural and widely used way to study information content in neuronal spike trains. As the reviewer noted, this indeed changes neuronal output properties at face value, because it is a non-linear transformation. For sparsely active neurons it will enhance the relative variability in z-score values for trials where the neuron shows low activity, but it will also similarly enhance the relative neural response when the neuron is in fact active. As highly active neurons (i.e. non-sparsely responding) are likely to also be highly variable (i.e. neurons generally have Fano factors higher than one), neural signals can often be better separated using normalized neuronal activity rather than absolute spiking rates or calcium fluorescence dF/F0 (Baddeley et al. 1997 as cited (subsection “Response dissimilarity within neuronal populations correlates with detection”, third paragraph); Montijn, Vinck, and Pennartz 2014). We agree with the reviewer that neuronal response variability and normalization procedures thereof are important and interesting topics to discuss, but we believe this is beyond the scope of our current manuscript (also considering the already substantial size of the manuscript). We trust that the clarifications of the heterogeneity calculation in combination with the additional analyses in the revised manuscript (Figure 7) will sufficiently address the reviewer’s concerns and better explain how to interpret heterogeneity, and its relevance for the brain.

*5) The alternative measure "instantaneous Pearson correlations" suffers from the same problems as "heterogeneity." It is improperly named as it is not a Pearson correlation. Time varying correlation measures already exist and should be mentioned; they are generally based on sliding windows (e.g. "Time-varying correlation coefficients estimation and its application to dynamic connectivity analysis of fMRI" Fu et al. 2013 or "The sliding window correlation procedure for detecting hidden correlations: existence of behavioral subgroups illustrated with aged rats" Schulz and Huston 2002).*

The initial name of “instantaneous Pearson correlation” might have been interpreted as being somewhat inaccurate, because these correlations are indeed not classical Pearson correlations, but are only computationally similar. We have therefore renamed them to “instantaneous Pearson-like correlations” in the revised manuscript. We believe this name is warranted, because they are instantaneous and share all underlying computational steps with Pearson correlations, except the last; they are both pairwise and rely on the relative distance from the mean in standard deviations. As the reviewer suggested, we have also added a sliding-window correlation analysis to our comparison of different metrics (Figure 4, Figure 3—figure supplement 2, subsection “Instantaneous Pearson-like correlations and sliding window Pearson correlations”). These correlations showed a similar performance as other metrics, and were significantly poorer at predicting reaction times than heterogeneity (Figure 4).

*6) The nature of the decoder used (subsection “Heterogeneity predicts reaction time”) is never explained in the main text or Methods. The extremely convoluted use of a similarity metric and p-value based on comparison to randomly shuffled data (Figure 4) to claim that the decoder and the animal behave similarly is not a clear and honest presentation of results. The similarity metric was not explained in the Methods. There is nothing to support the statement that "the performance as a function of contrast was strikingly similar to the animals' actual behavioral performance."*

In the original version of the manuscript we explained the basic decoding procedure and referenced to one of our previously published papers describing the nature of the decoder in extended detail (Montijn, Vinck, and Pennartz 2014). In the original version we also explained that the similarity metric was defined as the “Pearson correlation over contrasts” and was compared to a distribution of correlations after randomly shuffling over contrasts. We therefore believe our original statement was valid, honest and supported by our analyses, but we recognize the reviewer’s wish for further information on the specifics of the analyses. We have therefore revised the manuscript to explain in more detail our decoding procedure (subsection “Decoding of stimulus presence”), and have added an additional chi-square analysis of the trial-by-trial correspondence between the decoder’s output and the animals’ response (Figure 4—figure supplement 2, subsection “Heterogeneity predicts reaction time”, third paragraph and, subsection “Decoding of stimulus presence”, first paragraph). We have removed “strikingly” from the revised manuscript, as we agree with the reviewer this is somewhat subjective.

*7) The assertion, in the Introduction, that "a widely held assumption in computational models of vision is that neurons in distributed cortical architectures have relatively fixed roles in information coding" is a straw-man argument. The authors do not adequately characterize what this assumption of "fixed roles" means, and also fail to characterize the diverse set of existing theories and conjectures about how the visual system may function.*

We agree with the reviewer that the statement in the original version of our manuscript was insufficiently clear. We have edited the text to more accurately reflect our line of reasoning – that while the stimulus features represented by neurons are generally stable across time, neural modulations that regulate or affect whether physically identical stimuli are perceived may be more temporally dynamic (Introduction, last paragraph). As we also mention below, we have revised the manuscript to now refer to “population pattern consistencies” rather than “ensemble reoccurrences”.

*8) We need to see much more raw data so as to evaluate data quality. In particular, we should see supplementary movies showing simultaneous raw, unprocessed imaging data, behavior, and "heterogeneity" for ~10 consecutive trials.*

We have included a movie (Video 1) showing fluorescence data, the presented stimulus, licking responses, heterogeneity and mean population dF/F0 with the revised manuscript. Note that x-y shifts and z-drifts are small, both preceding and during stimulus presentation periods. During reward periods (after stimulus offsets) these shifts are more visible, but because we only take neural responses up to the first licking response, these spatial instabilities do not influence our neural data (see also below for a more comprehensive discussion of potential confounds related to z-shifts). Several neurons can be seen by eye to respond with increased fluorescence to visual stimulation.

*9) The very large responses of some neurons with nearly 100% DF/F in Figure 1 don't seem to match the very modest DF/F of 4% over "preferred populations" in Figure 2. Are the data in Figure 1 not representative of the full dataset? Or is the time window for averaging each trial's responses perhaps too long? The presentation, figure and analyses are unclear.*

As can be seen in Figure 2, in many trials the animal responds to the stimulus before the neuronal calcium response reaches maximum intensity. Moreover, because it takes time for the initial neuronal response to emerge in V1, and OGB’s rise time is a couple dozen milliseconds, the first couple of imaging frames yield baseline dF/F0 values (Chen et al. 2013). Averaging over only these early frames up to the animal’s response, where dF/F0 has not yet reached maximal intensity, therefore results in mean trial dF/F0 values that are much lower than the peak responses visible in the traces. As can also be seen in Figure 1 and Figure 2, during most trials some neurons respond vigorously, while others show very little change in dF/F0 (even within preferred populations), thereby further lowering the mean dF/F0 averaged across neurons.

*10) The first stated aim is to ask: “does visual detection correlate with mean visual response strength or other metrics?". This may be of interest if one could determine for certain that the response strength was being determined for the neurons really involved in the detection/perception required by the task. But why should we care what L2/3 is doing during this task, when it may not even be involved in generating the behavioral response?*

Although we have not performed experiments to test the causal involvement of mouse V1 in our specific task and setup, it has been reported before that (as we also mention in the first paragraph of the Introduction) mouse primary visual cortex is used to detect both orientation and contrast changes, as shown by optogenetic silencing of mostly superficial layers in V1 during behavioral task performance (Glickfeld et al., 2013). For our experiment stimulus orientation was irrelevant for obtaining reward, but the contrast-dependent stimulus detection in our design is similar to the one employed by Glickfeld and colleagues in their optogenetic study. It may therefore be reasonably assumed that the neuronal populations we recorded in L2/3 of mouse are causally involved in generating the behavioral response to visual stimuli with different contrasts. However, we agree with the reviewer that we were insufficiently clear in stating that our results only pertain to L2/3 neurons in mouse V1. We have changed the text in the manuscript so that it more accurately reflects the scope of our results (subsection “Possible cortical layer specificity of poor correlation of mean population responses “and throughout manuscript).

*11) The authors assert in the Introduction that "specific ensemble activation patterns reoccur across temporally spaced trials in association with hit responses, but not when the animal fails to report a stimulus." I do not understand how, on the basis of the data presented in Figure 6 and the manuscript text associated with it, that this conclusion can be drawn. The authors state: "We again split the data into miss, fast and slow response trials, and computed the correlations between response patterns from different trials separately for preferred and non-preferred neuronal populations…" What response patterns are being correlated? The Methods states that the "mean inter-trial correlations over animals" was compared. I find the link between this measure and the conventional definition of ensemble tenuous at best. Further, the calculated correlation coefficients are very low (<0.12), which does not support well the claim made above.*

In the original version when using “ensemble”, we meant to refer to the population of neurons we measured, conforming to the common use of the term "ensemble recordings" in the literature, but we agree that within certain disciplines “ensemble” would imply a stronger correlation of activation, similar to what “assembly” would entail. In the revised version of the manuscript we therefore now state that “neuronal populations show consistencies in activation patterns” (Introduction). Moreover, we now note that because we performed this analysis separately for preferred and non-preferred neurons, the low correlation values can be explained by a removal of much of the orientation-based signal in population responses (subsection “Population activation pattern consistency”). To further validate our results, we also performed as extra control a shuffling procedure. This control showed significantly lower correlation values than the unshuffled data for slow and fast, but not for miss trials (Figure 6).

*12) The authors describe their method for assessing the extent of slow drift in the z-plane, which they quantize into 10μm bins. It is unclear what additional effects this may have on the measured Ca^2^ -transients, something that would be best determined empirically using simultaneous electrophysiology. More importantly, fast shifts in the z-plane are a considerably larger problem, and these would be anticipated as the animal changes its posture or shifts fore- or hindlimb. This sort of "fidgeting" is commonly observed in advance of a rodent making a behavioural response. How the authors measured these postural adjustments is not clear, neither is the effect that these movements have on the activity recorded. It is certainly conceivable that a z-shift could move the focal plane further inside some neurons and further outside others, thereby increasing "heterogeneity."*

Although having addressed slow drifts in the original version of the manuscript, we had indeed not performed any quantification of fast shifts in z-plane, which – as the reviewer mentions – could present a major confound to our results. To address this issue in the revised version, we computed the depth in z-plane for each imaging frame. As we mention in the revised version of our manuscript, z-depth was quite stable (shifts rarely exceeded 1-2 microns) and pre-response z-shifts were not observable (subsections “Potential confounds” and “Z-drift quantification and recording stability analysis”). Some increase in z-shifts was visible following a behavioral hit response (Figure 1—figure supplement L-O), but these epochs were not included in any of our analyses, and can therefore not influence our results.

*13) Previous multiphoton Ca^2^ -imaging studies have shown that correcting xy-shifts uniformly across the whole image is not sufficient for motion correction in awake animals (see Dombeck et al. 2007, Greenberg and Kerr 2009). As described above, motion-associated artefacts resulting from the fidgeting of the animal around a response are not quantified and potentially important.*

We agree that motion artifacts can be a profound problem in calcium imaging experiments in awake animals. The first paper the reviewer mentions (Dombeck et al., 2007) shows that image warping within single frames occurs only during running, which limits the potential confound to running episodes. Nevertheless, within-frame image warping could indeed prove to be quite problematic when recording fluorescence data during locomotion. However, we believe that the noise levels induced by within-frame warping of the acquisition image as reported by Dombeck et al. (2007) and Greenberg and Kerr (2009) are most likely much higher than for our data sets for the following reasons. Dombeck et al. (2007) quantify the level of noise artifacts in an experimental setup where the mouse is running on a Styrofoam ball, rather than sitting on a treadmill. Mice placed on Styrofoam balls tend to run spontaneously because they are placed on a non-level surface, while all of our animals were trained for several months on a level-surface treadmill. Locomotion by our animals generally became less pronounced over several months of training, and had in all animals become quite rare during calcium imaging recordings.

More importantly, both studies mentioned by the reviewer used frame acquisition rates much lower than in our study (4 or 8 Hz by Dombeck et al., 2007 and 10.4 Hz by Greenberg and Kerr, 2009 vs. 25.4 Hz in our study). Within-frame image warping depends directly on the frame acquisition speed, because for low acquisition speeds there is more time for spatial movements in the brain to distort the image. As acquisition speed increases, brain tissue movement leads to displacement between frames rather than warping within frames. Such between-frame image displacement can still greatly influence data quality if left uncorrected, but these movements can be corrected using affine x-y registration algorithms, as we have done in our study (see our response to the previous comment for issues related to z-shifts and potential fidgeting behavior). We would also like to note that Greenberg and Kerr performed experiments on rats, where the dura was removed, which probably leads to significantly larger motion artifacts than in a mouse brain with intact dura. Still, these arguments provide only circumstantial proof that our data would show fewer and less strong motion artifacts. Therefore, to test for contamination by running-induced artifacts, we performed control analyses where we removed all trials during which the animals were running and found that this did not influence our results (Figure 2—figure supplement 1). Therefore the within-frame image warping artifacts the reviewer mentions are unlikely to influence our data to a significant degree (see also Video 1).

*14) The caveat that the only neurons from which recordings were made were superficial neurons ought to have been explicitly discussed. Is it not conceivable that the correlation of mean activity with perception might be significantly higher for neurons in deeper layers?*

We agree with the reviewer that this is an important issue. We have corrected this flaw in the revised version and now explicitly discuss that our observations relate only to L2/3, and that L5 might show a very different correlation with the detection of visual stimuli (subsection “Possible cortical layer specificity of poor correlation of mean population responses”).

*15) How did the authors control for possible ocular torsion (twisting of the eye and retina round the optic axis) during the experiment? This would totally invalidate all analyses based on orientation if present but not accounted for.*

Ocular torsion in mice has been reported to be less than in primates and humans, and rarely exceeds more than a couple of degrees (Mezey et al. 2004 (Vision Research 44); Goonetilleke et al. 2008 (Vision Research 48); Migliaccio, Meierhofer, and Santina 2010 (Experimental Brain Research 210)). We presented drifting gratings of 8 different directions to the mice, and thus they were spaced around the unit circle in steps of 45 degrees. The inter-stimulus difference of 45 degrees is therefore an order of magnitude higher than reported ocular torsions in mice. As such, it would have little impact on our analyses based on orientation. In any case, this potential confound would not influence our main result (the hit/miss and RT prediction of heterogeneity), because this does not depend on analyzing different orientations separately. Potential ocular torsion effects would influence our analyses based on explicit differential treatment of stimulus orientations (for example, the difference between preferred and non-preferred populations in Figure 3, and Figure 6), where its effect would not be to totally invalidate them, but to inject some extra noise into the results. We would also like to note that it is very uncommon to record ocular torsion in awake rodents, and the general consensus is that no corrections are necessary (e.g. Dombeck et al. 2007; Greenberg and Kerr 2009, as cited by the reviewer, also do not report any data on or potential confounds related to ocular torsion).

Reviewer #2:

*1) The concern is about the animal's behavior. The performances shown in Figure 1 are relatively low at 100% contrast; in many cases slightly different than the ones at 32%. The presence of errors at full contrast imply mechanisms other than visual detection contributing to the animal's response variability that will potentially contaminate all other conditions as well.* We understand the reviewer’s concerns about the animal’s performance, however, we believe that the behavioral mechanisms contributing to the animal’s decision to respond to 100% contrast visual stimuli are not reflected to a significant degree in the physiological hit/miss differences we find for test contrasts. As we mention in the original version of the manuscript, if V1 population dF/F0 or heterogeneity would be influenced by these factors, we would expect to also find a differential effect between hits and misses for 100% stimulus contrast trials. Our data show that this is not the case (Figure 3 and Figure 2—figure supplement 1); hit/miss differences in dF/F0 and heterogeneity are largest for intermediate contrasts. The animals’ suboptimal performance most likely dilutes the actual hit/miss difference, and therefore better behavioral performance would have probably led to larger, rather than smaller, effect sizes. We now also describe these caveats in the revised version of our manuscript (subsection “Potential confounds”).

*2) Regarding the correlation between heterogeneity and behavior, the authors claim that "…the increased spread of neuronal response strengths within a population determine the behavioral accuracy". This reviewer is concerned about how strong is the change in heterogeneity between hit and misses to support this claim. In his opinion the authors should explicitly quantify how predictive is the animal's decision from this population measure, on a trial-by-trial basis.*

We agree with the reviewer that providing a single-trial based analysis of linear separation of hits and misses would improve the quality of our manuscript. We have performed a single-trial based analysis of the predictability of the behavioral response with either mean dF/F0 or heterogeneity using a receiver operating characteristic (ROC) analysis (Figure 3; subsections “Response dissimilarity within neuronal populations correlates with detection”, last paragraph and “Receiver Operating Characteristic (ROC) analysis of hit/miss separabilit”). As we mention in the manuscript, single trial linear predictions were significantly above chance for both dF/F0 and heterogeneity, and heterogeneity performed significantly better than dF/F0.

*3) He finds very interesting the fact that the measure of heterogeneity – but not the mean population response – correlates with detection. However, as far as he understands, this would be the case in any situation in which the detection of the stimulus is represented by a population code that is not merely an increase of activity of all neurons. The mean population response is only one particular projection of the population activity (let's say, described by the vector [1 1* … *1]). If detecting the stimulus activates the neural population in any other direction in neural space, this measure of heterogeneity will increase (because some neurons increase activity while others decrease). In particular, if detecting or not the stimulus modulates the population activity in a direction orthogonal to [1 1 … 1], the mean population response will not be affected (and won't correlate with the animal's behavior). His concern is that, if this is the case, it is not heterogeneity per se that is relevant, but the presence of complex population patterns of activity that are not visible at the level of the mean response. He thinks the authors should check if there is a population signal other than the mean response that correlates with the animal's decision.*

We agree with the reviewer this is a very interesting question and thank him/her for this excellent suggestion. It has inspired us to perform extra analyses that we describe in the revised version of our manuscript (Figure 7, subsection “Analysis of heterogeneity in multidimensional space” and “Analysis of multidimensional inter-trial distance in neural activity”). In summary, in order to study multidimensional aspects of population responses and their relation to heterogeneity, we had to develop an alternative definition of multidimensional heterogeneity. We argue that within a multidimensional neural response space (where each neuron represents a single dimension), heterogeneity within a population response corresponds to the distance from this response to the main diagonal, which represents the gradient along which the mean of the response changes. Here, the mean and heterogeneity are two complementary features of population responses with gradients in orthogonal directions (Figure 7). We studied whether population responses were distributed symmetrically around the main diagonal (i.e. no complex multidimensional patterns are present, other than captured by heterogeneity) or rather asymmetrically, which would show that heterogeneity is most likely an epiphenomenon. Population responses showed a modest, but significant asymmetry, for hit as well as miss trials, but interestingly hits were more asymmetrically distributed than misses (Figure 7). We provide further analyses where we removed either the mean or the heterogeneity of population responses (but preserved all other response structures) and quantify the effect of their removal on decoding the animal’s behavioral response. Removal of the mean had little effect, but removal of heterogeneity significantly impaired decoding performance (Figure 7). We thus can confirm that heterogeneity is an important factor that is contributing more to the differentiation between detected and non-detected stimuli than the mean of the population response, but that other, more complex population response properties contribute even more to the detection of visual stimuli than heterogeneity. We describe these results in more detail in the revised manuscript (subsection “Analysis of heterogeneity in multidimensional space”).

*4) The authors claim that ensemble patterns reoccur upon presentation of the same stimulus. However, inter-trial correlations of population responses are relatively low (~0.11). They should explain what value they take as a reference to validate this claim and why. Correlations could increase because of reasons other than reoccurring of the same activity pattern; a more detailed analysis is needed to support this claim.*

In the revised manuscript we now describe in more detail the exact procedures of calculating the population activity pattern similarities, and note that the low correlations can be explained by the separate analysis of preferred and non-preferred populations, which removes pattern correlations due to stimulus tuning (subsection “Population activation pattern consistency”). Moreover, we performed an additional control analysis where we shuffled trials randomly across neurons, but kept the stimulus types intact. This way, we kept the stimulus response properties the same for each individual neuron, but destroyed temporal inter- relations between the activity of neurons. This control revealed that our analyses indeed have an inherent bias towards small, non-zero correlations. Interestingly, shuffled correlations were almost identical to correlations during miss trials, but were significantly lower than non-shuffled correlations during fast and slow responses (Figure 6).

*5) He believes it is necessary to explain why the authors chose this particular measure of spread in neural responses, as opposed to – arguably – more natural ones like the variance. If the variance does not correlate with behavior as much as heterogeneity does, then this might also be informative of the properties of the population code. A set of related statistics are examined in regard to reaction times (Figure 4) but not in relation to the decision of the animal.*

As we mentioned in response to this reviewer’s third comment, we have significantly extended our manuscript with an in-depth analysis of the nature of heterogeneity in neuronal population responses (Figure 7, subsection “Analysis of heterogeneity in multidimensional space”). As the reviewer requested, we have also performed similar analyses of effect size for hits and misses on other measures (Figure 3—figure supplement 2). These analyses showed that heterogeneity performed significantly better than mean-based measures, but was not statistically different from other metrics (such the variance or correlation-based measures) in distinguishing hits and misses. We edited the revised manuscript accordingly to more accurately reflect that heterogeneity is specifically better than other measures in explaining reaction time variability, but only better than mean-based metrics in hit/miss differentiation.

Reviewer #3:

Negative points:1) Preferred orientation and non-preferred orientation neurons are analyzed separately – this ignores potential interactions between neurons (subsection “Data processing”).

As we also mentioned in response to the first comment by reviewer 1, we analyzed preferred and non- preferred populations only separately for Figure 2 and Figure 6. We apologize for the apparently ambiguous explanation in the original version of the manuscript and have attempted to more clearly state in the revised version that we include all neurons in almost all analyses (subsection “Response dissimilarity within neuronal populations correlates with detection”, first, second and fourth paragraphs).

*2) The preferred orientation neurons are selected using the mean dF/F0 value, however, the main result of the paper suggests that a different metric, heterogeneity, is more robust in capturing stimuli recognition; how will the analysis be affected if the same metric is used for pruning the neurons? (subsection “Calculation of preferred stimulus orientation”*).

As heterogeneity is a population-based metric, it is ill-suited to be applied to single neurons. Only a jackknifing procedure would allow an estimation of the effect single neurons have on population heterogeneity on average. However, even in this case a neuron’s contribution to population heterogeneity would depend on the activation of other neurons (see Eqs. 2-4), and therefore does not represent a unique feature of single neurons. However, we recognize the reviewer’s concern about potentially different effects occurring in preferred and non-preferred populations. We have therefore performed an analysis of the hit/miss effect size in dF/F0 and heterogeneity for the preferred, non-preferred and whole populations (Figure 3 and Figure 3—figure supplement 1. Also note that Figure 4—figure supplement 2 shows that heterogeneity differences exist between hits and misses in all quintiles of neurons when sorted by relative activity, further supporting our results that behavioral correlates of heterogeneity are distributed among the entire population, including weakly active and non-preferred neurons.

*3) As defined, heterogeneity seems a reasonable metric, however, it only considers pairwise relationships between neurons; a more holistic, group-level metric should be considered, since the goal of the analysis is to discover groups of neurons.*

We agree with the reviewer that pairwise relationships between neurons are a narrow spectrum of interactions to consider. We have therefore added extensive new analyses to the revised version of the manuscript, including an alternative, 'holistic' definition of heterogeneity (Figure 7 and subsections “Analysis of heterogeneity in multidimensional space, “Analysis of multidimensional inter-trial distance in neural activity”, and “Removal of mean and heterogeneity, and subsequent hit/miss decoding”). As we note in the revised manuscript, our analyses suggest that the differentiation in heterogeneity between hits and misses is at least in part non-epiphenomenal, but that detection vs. non-detection differences also reside in neuronal population response patterns other than its mean or heterogeneity (subsections “Analysis of heterogeneity in multidimensional space” and “Multidimensional analysis: heterogeneity contributes to, but does not fully account for visual detection”).

*4) Can you explain or cite the reasoning behind using the procedure in the subsection “Behavioral response predictability on single trial basis”, to compute a prediction? Can the model likelihood be used to make predictions instead?*

The procedure the reviewer refers to erroneously stated in the original version that the calculation was based on the relative decoding probabilities of response types (miss, fast, or slow), while in fact it was already the relative likelihood of response types that determined the output. We have corrected this error in the revised version of the manuscript (subsection “Behavioral response predictability on single trial basis”). We would also like to note that in the revised version we have now performed an additional single-trial based hit/miss analysis using receiver operating characteristic (ROC) curves (Figure 3). We hope this sufficiently addresses the reviewer’s concerns.

[Editors' note: further revisions were requested prior to acceptance, as described below.]

Reviewer #1:

*Regarding the authors’ response to Reviewer #1, comment 10: The study by Glickfeld activated PV neurons in a 1mm diameter around the injection pipettes ~up to 1mm below the V1 surface and showed that this increased the threshold for detection of both orientation and contrast by the animals. I do not see the relationship between the author's response to my question and the question. Why is it reasonably assumed that L2/3 is involved in the task that is presented?* We agree with the reviewer, as we also stated in our initial reply, that we cannot claim to have shown causal involvement of V1 L2/3 neurons in detecting stimuli. In our manuscript we therefore only discuss our results in terms of correlation, rather than causation. However, the results presented by Glickfeld and colleagues also show that their effects of optogenetic stimulation were stronger for superficial than deep layers (reduction of firing rate: -66.7% for units in superficial layers, vs. -33.9% for units in deep layers), as the channelrhodopsin-exciting blue light was shone unto the cortical surface and was therefore less intense when it reached deeper layers. Arguably, the reduction in behavioral performance could therefore be due mostly to the stronger neural effect in superficial layers. This is of course hypothetical, as Glickfeld et al. did not explicitly quantify the dependence of their behavioral effect on the reduction in firing rate in superficial vs. deep layers. However, we believe that correlational rather than causational research is still valuable in itself, as it may provide new insights and ideas on how the visual cortex could be involved in the detection of visual stimuli. Moreover, it has been reported that superficial layers (L2/3) in visual cortex correlate with the perception of visual stimuli (e.g. Ito and Gilbert 1999; van der Togt et al., 2006). In summary, although there is no proof yet for the causal involvement of L2/3 in visual detection, we believe these considerations offer sufficient justification for examining L2/3 population behavior. We have added references to these previous studies that show correlates in superficial layers to the revised manuscript (Introduction, last paragraph).

*Regarding the authors’ response to Reviewer #1, comment 11: Please change the last sentence in the Abstract to reflect the changes in terminology by removing ensembles (see below). I’m not sure what “selective and dynamic neuronal ensembles” are. Please also rephrase the first paragraph of the Discussion, which suffers from the same issue.*

From the Abstract:"Contrary to models relying on temporally stable networks or bulk-signaling, these results suggest that detection depends on transient activation of selective and dynamic neuronal ensembles."

We agree with the reviewer and have changed the corresponding sentence to “detection depends on transient differentiation in neuronal activity within cortical populations”, which more accurately reflects the main findings of our study (Abstract and Discussion, first paragraph). We have also removed our reference to ensemble formation from the Discussion.

Reviewer #2:

*1) I had noted that the low performances at full contrast imply mechanisms other than visual detection contributing to the animal's decision (lack of motivation, for example). This means that test contrast trials are probably contaminated with a significant amount of trials (close to 50% for several animals) in which the animal actually detected the stimulus but didn't respond. The authors argue that heterogeneity does not reflect these other mechanisms because it's equal for both behavioral responses at full contrast. I agree with the argument and understand that the low performances might actually be diluting the effect reported in the paper. But I still would like to ask, does the distribution of heterogeneity in "No Resp" trials show any hint of bimodality, reflecting the 50% of trials in which the stimulus was in fact detected?*

A coarse analysis of the distribution of heterogeneity values across miss trials during test contrasts shows that there might be a slight bimodality (or at least, a skewed distribution; see below). Supposing the distribution of heterogeneity values during miss trials is indeed bimodal, the effect size between the two underlying distributions (Cohen’s d) would be 0.9641 (difference in standard deviations between red and blue curves in Figure 8). However, this is of course hypothetical, as the distribution shown may very well be unimodal. Given the current data, and considering that testing for bimodality is a notoriously difficult problem, it is therefore hard to draw a strong conclusion. Because this analysis gave mostly inconclusive results, we have not incorporated these findings in the manuscript.

Author response image 1.**DOI:**
http://dx.doi.org/10.7554/eLife.10163.018